# Deep generative model embedding of single-cell RNA-Seq profiles on hyperspheres and hyperbolic spaces

Jiarui Ding [1✉] & Aviv Regev [1,2,3✉]

Single-cell RNA-Seq (scRNA-seq) is invaluable for studying biological systems. Dimensionality reduction is a crucial step in interpreting the relation between cells in scRNA-seq data. However, current dimensionality reduction methods are often confounded by multiple simultaneous technical and biological variability, result in "crowding" of cells in the center of the latent space, or inadequately capture temporal relationships. Here, we introduce scPhere, a scalable deep generative model to embed cells into low-dimensional hyperspherical or hyperbolic spaces to accurately represent scRNA-seq data. ScPhere addresses multi-level, complex batch factors, facilitates the interactive visualization of large datasets, resolves cell crowding, and uncovers temporal trajectories. We demonstrate scPhere on nine large datasets in complex tissue from human patients or animal development. Our results show how scPhere facilitates the interpretation of scRNA-seq data by generating batch-invariant embeddings to map data from new individuals, identifies cell types affected by biological variables, infers cells' spatial positions in pre-defined biological specimens, and highlights complex cellular relations.

[1] Klarman Cell Observatory, Broad Institute of MIT and Harvard, Cambridge, MA, USA. [2] Howard Hughes Medical Institute, Koch Institute of Integrative Cancer Research, Department of Biology, Massachusetts Institute of Technology, Cambridge, MA, USA. [3] Present address: Genentech, South San Francisco, CA, USA. ✉email: jding@broadinstitute.org; aviv.regev.sc@gmail.com

Single-cell genomics—especially single-cell RNA-seq (scRNA-seq)—has opened the way to a comprehensive analysis of the relationship between cells, including their different types, states, physiological transitions, differentiation trajectories, and spatial positions[1–3]. Although scRNA-seq datasets have high dimensionality, their intrinsic dimensionality is typically low, because many genes are co-expressed and a few variables, such as cell type, a gene program, or the number of detected transcripts, could explain a substantial portion of the variation in a dataset. As a result, dimensionality reduction, followed by visualization or downstream analyses has become a key strategy for exploratory data analysis in single-cell genomics[4,5].

Recently, deep-learning models[6], especially (variational) autoencoders[7–9], have been used for dimensionality reduction prior to visualization or downstream analyses, such as clustering[10–15]. This leverages their ability to model large-scale high-dimensional data and their flexibility in incorporating different factors, especially batch effects in the modeling framework. Moreover, such models can provide an end-to-end, single process for analyses that otherwise require multiple separate steps, each with its own method or algorithm, including batch correction, dimensionality reduction, and visualization.

However, standard variational autoencoders (VAEs) have several shortcomings when modeling and analyzing scRNA-seq data. First, they assume a multidimensional normal prior for the low-dimensional latent variables, which unfortunately encourages the low-dimensional representations of all cells to the group in the center of the latent space, even for data consisting of distinct cell types. This is especially true if the model is trained long enough, such that the posterior distributions gradually approximate the prior distribution. (Cell crowding also afflicts general-purpose data visualization tools such as *t*-stochastic neighborhood embedding (*t*-SNE)[16], once the large datasets consist of hundreds of thousands of cells[17,18].) A second challenge arises from using the cosine to measure the distance between two cells[19–21] for very sparse droplet-based scRNA-seq data (>90% genes with zero counts in a typical cell profile). Because the cosine distance between two cell vectors is their Euclidean distance after normalizing the two cell vectors to have a unit $\ell^2$ norm, the cells lie on the surface of a unit hypersphere with a dimensionality of $D-1$, where $D$ is the number of measured genes. Embedding data distributed on a hypersphere to a Euclidean space introduces significant distortion for commonly used dimensionality reduction tools[22], and standard variational autoencoders also fail to model such data[23]. Moreover, the Euclidean geometry is not optimal for representing hierarchical, branched developmental trajectories[24–26]. Third, in practice, current applications of VAEs for scRNA-seq data can only handle a single-batch vector (factor), whereas biologically relevant datasets typically have multiple such factors, both technical (e.g., replicate or study) and biological (e.g., patient, tissue location, disease status). Such complex multilevel factors are not well-handled by current batch-correction methods in single-cell genomics, either VAEs or other approaches[5,12,27–31], but addressing them is critical for integration across studies, interpretation of the impact of various factors on cells in complex tissues, and the ultimate assembly of large tissue atlases.

Here, we present alternative approaches for embedding of cells into hyperspherical or hyperbolic spaces based on deep-generative models, to better capture their inherent properties, tackle complex batch effects, generate references, and perform diverse analyses. For general scRNA-seq data, we minimize the distortion by embedding cells to a lower-dimensional hypersphere instead of a low-dimensional Euclidean space[23], using von Mises–Fisher (vMF) distributions on hyperspheres as the posteriors for the latent variables[23,32,33]. Because the prior is a uniform distribution on a unit hypersphere and the uniform distribution on a hypersphere has no centers, points are no longer forced to cluster in the center of the latent space. For representation and inference of hierarchical, branched developmental trajectories, we embed cells to the hyperbolic space of the Lorentz model and visualize the embedding in a Poincaré disk[24,25,34]. Using nine diverse datasets from human and model organisms, we demonstrate scPhere's superior performance on key existing use cases as well as emerging applications, including processing large scRNA-seq datasets with complex multilevel batch effects, visualizing cell profiles from highly complex tissues and developmental processes, building batch-invariant reference models to which new data can be readily mapped, identifying the cells impacted by specific biological factors, and mapping cells to spatial positions. Overall, our model provides enhanced representation, complex batch correction, reference-generation, visualization, and an interpretation tool for single-cell genomics research.

## Results

**Mapping scRNA-seq data to hyperspherical or hyperbolic latent spaces**. We developed scPhere (pronounced "sphere"), a deep-learning method that takes scRNA-seq count data and information about multiple known confounding factors (e.g., batches, conditions) and embeds the cells to a hyperspherical or hyperbolic latent space (Fig. 1a, "Methods"). We reasoned that scPhere would allow cells to be embedded more appropriately because they will not be constrained to aggregate in the center. In cases where we expect a branching structure with a large number of trajectories, hyperbolic spaces are particularly suitable, because the exponential volume growth of hyperbolic spaces with radius confers them enough capacity to embed trees, which have exponentially increasing numbers of nodes with depth. For 3D visualization, scPhere places cells on the surface area of a sphere (but not inside the sphere), such that we only need to rotate the sphere to see all cells, avoiding the common challenge of exploring the interior of 3D embeddings. The scPhere package renders all 3D plots for interactive visualizations of millions of cells with the rapid rgl R package, with web graphics library files, which can be opened in a browser for exploration. Alternatively, one can convert the 3D coordinates to 2D, based on various projection methods, such as the recent Equal Earth map projection method[35].

Specifically, scPhere takes as input an scRNA-seq dataset $D = \{(\mathbf{x}_i, \mathbf{y}_i)\}_{i=1}^{N}$ with $N$ cells, where $\mathbf{x}_i$ is the UMI count vector of $D$ genes in cell $i$, and $\mathbf{y}_i$ is a categorical vector specifying the batch in which $\mathbf{x}_i$ is measured, and models the $\mathbf{x}_i$ UMI count distribution as governed by a latent low-dimensional random vector $\mathbf{z}_i$ and by $\mathbf{y}_i$ (Fig. 1a, "Methods"). Note that $\mathbf{y}_i$ can account for multilevel confounding factors, for example, patient, disease status, and lab protocol. The scPhere model assumes that the latent low-dimensional random vector $\mathbf{z}_i$ is distributed according to a prior, with the joint distribution of the whole model factored as $p(\mathbf{y}_i|\boldsymbol{\theta}_i)p(\mathbf{z}_i|\boldsymbol{\theta}_i)p(\mathbf{x}_i|\mathbf{y}_i,\mathbf{z}_i,\boldsymbol{\theta}_i)$, where $p(\mathbf{y}_i|\boldsymbol{\theta}_i)$ is the categorical probability mass function (constant for our case, as $\mathbf{y}_i$ is observed). For hyperspherical latent spaces, scPhere uses a uniform prior on a hypersphere for $p(\mathbf{z}_i|\boldsymbol{\theta}_i)$; for hyperbolic latent spaces, it uses a wrapped normal distribution in the hyperbolic space as the prior. For the observed raw UMI count inputs, we assume a negative-binomial distribution: $p(\mathbf{x}_i|\mathbf{y}_i, \mathbf{z}_i, \boldsymbol{\theta}_i) = \prod_{j=1}^{D} \mathrm{NB}(x_{i,j}|\mu_{\mathbf{y}_i,\mathbf{z}_i}, \sigma_{\mathbf{y}_i,\mathbf{z}_i})$, with parameters specified by a neural network. The inference problem is to compute the posterior distribution $p(\mathbf{z}_i|\mathbf{y}_i, \mathbf{x}_i, \boldsymbol{\theta}_i)$, which is assumed to be a von Mises–Fisher distribution for hyperspherical latent spaces, and a

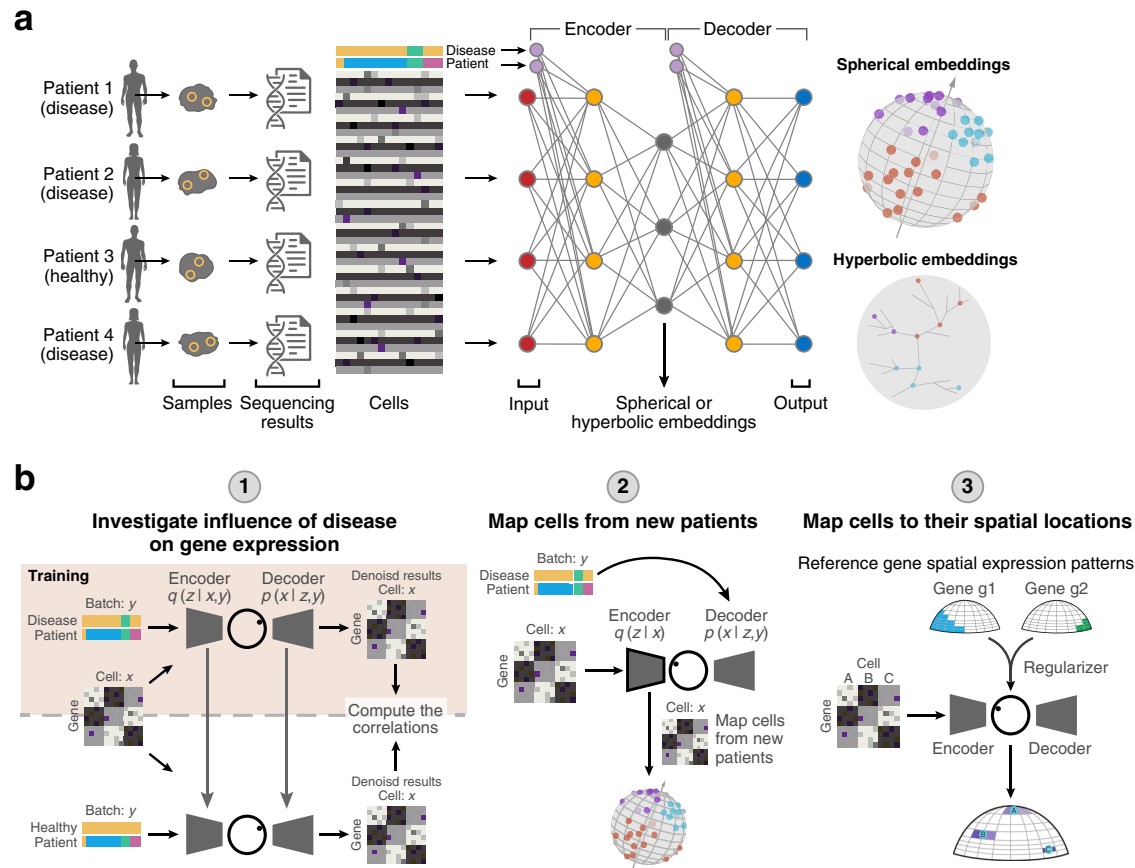

**Fig. 1 ScPhere model. a**, **b** Method overview. **a** ScPhere takes as input scRNA-seq measurements and multilevel technical or biological batch effects (e.g., replicate patient, disease) and learns cells' latent structure accounting for the batch effects. **b** The learned scPhere model and the low-dimensional latent representations of cells are used for diverse queries, including (1) identifying the influence of a biological factor on expression, (2) generating a batch-invariant reference and mapping new datasets to the reference, and (3) mapping cells to their spatial locations.

wrapped normal distribution for hyperbolic latent spaces. Because it is intractable to compute the posterior, the scPhere model uses a variational distribution $q(\mathbf{z}_i | \mathbf{y}_i, \mathbf{x}_i, \boldsymbol{\phi}_i)$ to approximate the posterior (Fig. 1a, "Methods"). When a hyperspherical latent space is used, $\mathbf{x}_i$ is first log-transformed and scaled to have a unit $\ell^2$ norm for inference, otherwise $\mathbf{x}_i$ is only log-transformed but not scaled. The parameters $\boldsymbol{\phi}_i$ of the variational distribution are (continuous) functions of $\mathbf{x}_i$ and $\mathbf{y}_i$ parameterized by a neural network with parameter $\boldsymbol{\phi}$. As a deep-learning model trained by mini-batch stochastic gradient descent, scPhere is especially suited to process large scRNA-seq datasets with complex multilevel batch effects and facilitates emerging applications (Fig. 1b). We provide full details in the "Methods" section.

**ScPhere visualizes large datasets with multiple cell types and hierarchical structures without cell crowding.** Applying scPhere to scRNA-seq data shows that its spherical latent variables help address the problem of cell crowding in the origin and that it provides excellent visualization for data exploration, with easily interpretable latent variable posterior means of cells.

To illustrate this, we applied scPhere to six scRNA-seq datasets from human and mouse, spanning from small (thousands) to very large (hundreds of thousands) of cells from one or multiple tissues, and with a small (two) to very large (dozens) of expected cell types. We compared scPhere's visualization with a hyperspherical latent space to scPhere's VAE but with a Euclidean embedding, as well as to three major general-purpose data visualization tools commonly applied to scRNA-seq data:

$t$-SNE[16], UMAP[36], and PHATE[37]. The "small" datasets were: (1) a blood cell dataset[38] with only 10 erythroid cell profiles and 2293 CD14$^+$ monocytes; (2) 3314 human lung cells[39], (3) 1378 mouse white adipose tissue stromal cells[40], and (4) 1755 human splenic nature killer cells spanning four subtypes[41]. The "large" datasets were: (1) 35,699 retinal ganglion cells in 45 cell subsets[42]; and (2) 599,926 cells spanning 102 subsets across 59 human tissues in the Human Cell Landscape[43].

Applying scPhere with a hyperspherical latent space to each of the "small" datasets readily distinguished cell subsets, and moreover, the posterior means of cells typically did not overlap, which helped ensure that we can discern individual cells without occlusion. In each case, cells of the same type were close to each other on the surface of a sphere, and yet generally two cells were distinguishable, even by eye (Supplementary Fig. 1a–d). Conversely, when we used a standard multivariate normal prior, the posterior means of the latent variables were centered at the origin, leading to crowding (Supplementary Fig. 1e–h). Thus, in the Euclidean space, the closer the cells were to the center, the higher were their densities, a problem persisting in both 2D (Supplementary Fig. 1e–h) and 3D (Supplementary Fig. 1i–l), even with rotation of the 3D space. In particular, similar cell types were very close to each other in the Euclidean space (e.g., APC and FIP, Supplementary Fig. 1g), and rare cell types became "outliers" (hNK_Sp3 and hNK_Sp4, Supplementary Fig. 1h). Notably, although there were discrete cell types in these datasets, even scPhere with hyperbolic latent spaces performed well (Supplementary Fig. 2a). Overall, $t$-SNE, UMAP, and PHATE[37] generally worked well for these smaller datasets without batch effects

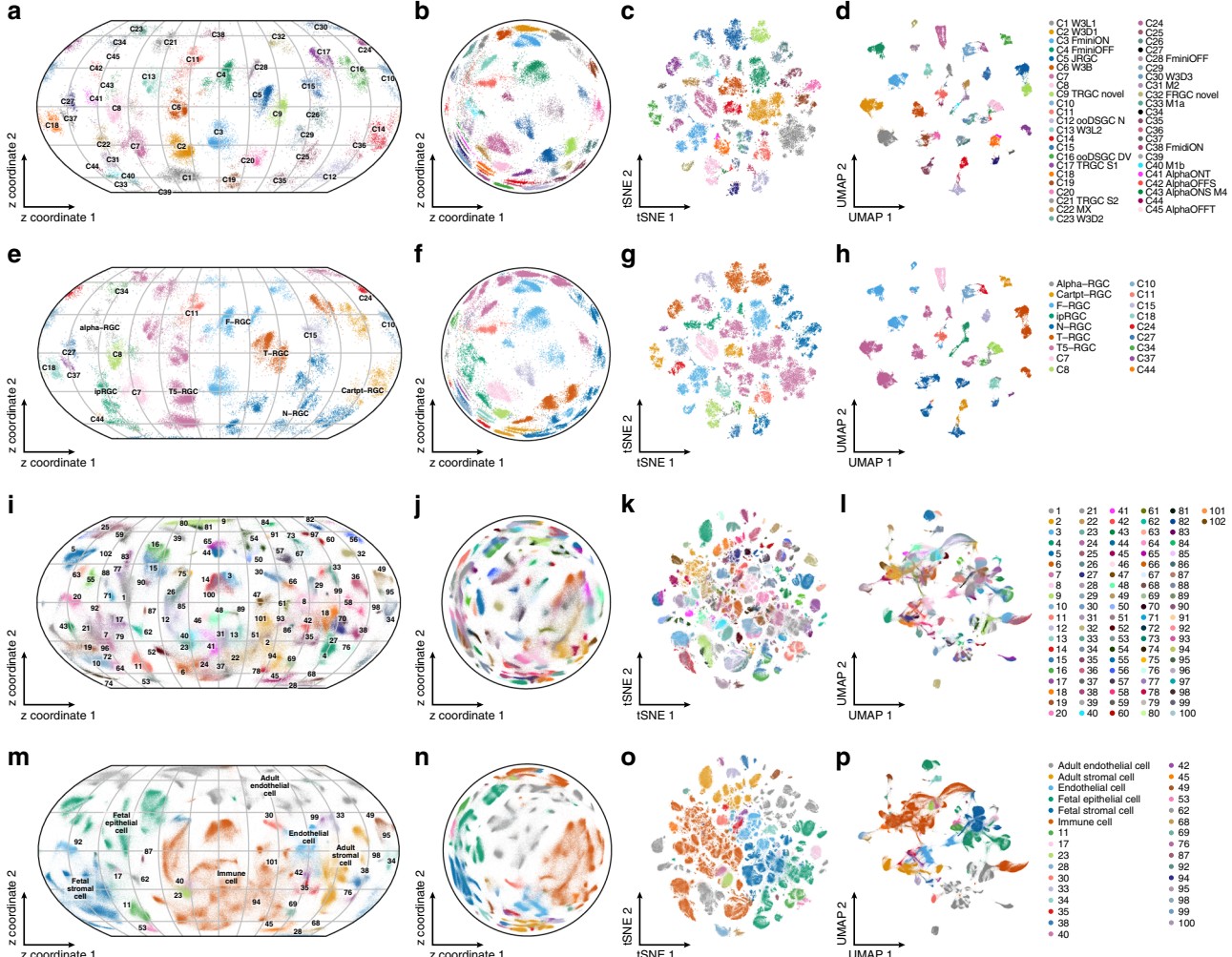

**Fig. 2 ScPhere resolves the cell-crowding problem and preserves hierarchical global structures in visualizing large scRNA-seq data. a–p** ScPhere learns latent representations that provide excellent visualization of local and global structure, even in very large datasets. Shown are scPhere posterior means of cells in a sphere projected to 2D by the Equal Earth map projection method (**a, e, i, m**), scPhere learned representations in the Poincaré disk (**b, f, j, n**), 2D t-stochastic neighborhood embeddings (t-SNE) (**c, g, k, o**), and 2D uniform manifold approximation and projections (UMAP) (**d, h, l, p**), for 45 mouse retinal ganglion cells (RGCs) (**a–h**), and 102 Human Cell Landscape clusters (**i–p**) with cells colored by either cluster membership (**a–d, i–l**), or by major cell types (**e–h, m–p**; clusters that cannot be assigned to major cell types are numbered).

(Supplementary Fig. 2b–d), with some minor challenges (e.g., mixing of mouse adipose doublets and macrophages by UMAP, Supplementary Fig. 2c), and PHATE—designed for development trajectories—connecting cells inaccurately when only discrete cell types are present (Supplementary Fig. 2d).

ScPhere's advantages compared to other approaches were particularly pronounced when applied to datasets with a larger number of cells and clusters: mouse retinal ganglion cells (RGCs)[42] and the Human Cell Landscape[43]. While scPhere (with either spherical or hyperbolic latent space and default parameters throughout), t-SNE and UMAP all discerned well individual cell types among RGCs (Fig. 2a–h and Supplementary Fig. 3a–c) and the Human Cell Landscape (Fig. 2i–p and Supplementary Fig. 3d–f), scPhere best preserved the hierarchical global structure in these data, grouping together sets of clusters of different subtypes of each major type (Fig. 2e–h, m–p). For example, among RGCs, all Cartpt-RGC clusters were in one part of the scPhere embedding (Fig. 2e, f), but in different parts of the t-SNE and UMAP embeddings (Fig. 2g, h). Similarly, most of the 102-cell clusters in the Human Cell Landscape organize in the scPhere embedding by their six major cell groups (fetal stromal cells, fetal epithelial cells, adult endothelial cells, endothelial cells, adult

stromal cells, and immune cells) (Fig. 2m, n), but were more spread in different parts of the t-SNE and UMAP representations (Fig. 2o, p). ScPhere outperformed other methods in preserving the hierarchical global structure, based on global k-NN accuracies on both the RGC and the HCL datasets (scPhere: 73.53% and 92.21%, t-SNE: 47.06% and 79.22%; UMAP:52.94% and 83.12%, Supplementary Fig. 3b, e, "Methods"). Moreover, with this large number of cells, t-SNEs were increasingly "crowded"[17,18], such that where even very distinct cell types were very close to each other in 2D (Fig. 2k), and cells from multiple clusters appeared mixed in the UMAP (Fig. 2l), as reflected both visually and based on mean Silhouette scores (Mann–Whitney U test, FDR < 0.0001, Supplementary Fig. 3f, "Methods"). ScPhere did not suffer from these problems, partially because it was trained using mini-batches, while t-SNE and UMAP were learned using all the data, and their parameters (especially the perplexity parameter of t-SNE) have to be adapted to this larger number of cells, but increasing the perplexity parameter makes t-SNE computationally expensive[18]. (In our analyses, we used a list of perplexity parameters that already grow with the number of cells[18]; the cell-crowding problem suggests that much larger perplexity parameters are required.) By contrast, scPhere was trained with

default parameters and is scalable to process a large number of cells, with a time complexity that is linear with the number of input cells (Supplementary Fig. 4a–i, "Methods"). Embedding cells into the Euclidean space performed worse than embedding cells into hyperspherical latent spaces in terms of discerning discrete cell types or in preserving their hierarchical organization (Supplementary Fig. 3g–j), because the normal prior encourages the posterior means of cells to be centered at the origin. As expected, PHATE did not perform well for these large datasets of mostly discrete cell types (Supplementary Fig. 3k–n).

**ScPhere effectively models complex, multilevel batch, and other variables.** Single-cell profiles in realistic biological datasets are typically impacted by diverse factors, including technical batch effects in separate experiments and different lab protocols, as well as biological factors, such as interindividual variation, sex, disease, or tissue location. However, most batch-correction methods[5,12,27–29,31] handle only one batch variable (which often is technical in practice) and may not be well-suited to the increasing complexity of current datasets. ScPhere however can learn models of data with multiple variables.

To assess its ability to performed batch correction, we applied scPhere to a dataset of 301,749 cells we previously profiled in a complex experimental design from the colon mucosa of 18 patients with ulcerative colitis (UC), a major type of inflammatory bowel diseases (IBD), and 12 healthy individuals[44]. In addition to each individual patient biopsy being a batch, there were many other factors to consider: individuals were either healthy or with UC, cells were collected separately from the epithelial and lamina propria fractions of each biopsy, there were two replicate biopsies for each healthy individual and as a pair of inflamed and uninflamed biopsies for the UC patients (for a few UC patients, there were replicate inflamed and/or replicate uninflamed biopsies)[44], and, finally, samples were collected at two time periods, separated by over a year (analyzed as train and test data in the original study[44]). Notably, these factors had a substantial impact on the cells' profiles and ability to integrate the data, which required a large number of dedicated and iterative steps in the original study[44], with optimization for the specific dataset. To test scPhere, we applied it with default parameters, in a single end-to-end process, assessed its results biologically, and compared its performance to that of three leading batch-correction methods—Harmony[30], LIGER[29], and Seurat3 CCA[5] (the latter two can handle only one batch factor, which we chose to be the individual[44], as is the common practice; "Methods").

Analyzing cells with the patient origin as the batch vector, not only recapitulated the main cell groups in our initial study[44] but was highly refined, allowing us to better visually explore cellular relations (Fig. 3a–c and Supplementary Movies 1–4). For example, in the stromal and glial cells, endothelial cells and microvascular cells were close to each other, and adjacent to postcapillary venules. Conversely, these distinctions can barely be discerned in a UMAP plot of the same data, where endothelial and microvascular cells were very close (Supplementary Fig. 5a; using the 20 batch-corrected components by either Harmony[30], Seurat3 CCA[5], or LIGER[29] as inputs). Among fibroblasts, cells arranged in a manner that mirrored their position along the crypt–villus axis, from $RSPO3^+WNT2B^+$ cells (which support the ISC niche[44]), to $WNT2B^+$ cells, to $WNT5B^+$ cells. Strikingly, the inflammatory fibroblasts, which are unique to UC patients[44], were readily visible (Fig. 3a, light blue), and were both distinctive from the other fibroblasts, while spanning the range of the "crypt–villus axis" (as shown experimentally[44]). ScPhere's batch correction on this complex dataset (30 patients with disease and location factors) performed better than Harmony, Seurat3 CCA,

and LIGER based on classification accuracies of cell types for stromal, epithelial, and immune cells (Fig. 3d–f, Supplementary Figs. 6–9, either $k$-nearest neighbors ($k$-NN) or logistic regression; we omitted Seurat3 CCA results for immune cells with >200,000 cells and 30 batches, as it failed to complete.). ScPhere performed well even when using fewer latent variables, which avoids the component-collapse problem in VAEs (Supplementary Fig. 10, "Methods").

ScPhere's ability to correct for multiple confounding factors simultaneously (which is not readily possible with many other batch-correction methods[5,12,27–29]) helps to understand the impact of biological factors. For example, when using both patient origin and disease status (healthy, uninflamed, inflamed) as the batch vector in the stromal cells, scPhere largely merged the inflammatory fibroblasts with $WNT2B^+$ fibroblasts (Fig. 3g and Supplementary Movie 2). When analyzing epithelial cells, adding anatomical regions as a component of the batch vector, the cells were grouped solely by types (e.g., stem cells separate from TA2 cells, Fig. 3h, i), whereas anatomical regions dominated the cells, which organized in two respective parallel tracts in some regions of the sphere (Fig. 3j, k). Cell types that were mostly from one region (e.g., tuft cells, mostly from epithelial fractions) remained grouped distinctly (Fig. 3j, k). Similarly, when we did not use disease status (healthy, uninflamed, or inflamed) as a component of the batch vector, some cell types (e.g., TA2, immature enterocytes, and enterocytes) had "outliers" mapped to low-density regions of the sphere (Fig. 3l), mostly from UC samples (Fig. 3m), but the cells formed more compact clusters once disease status was included (Supplementary Fig. 5b), with good mixing between the cells from different disease states and patients (Supplementary Fig. 5c, d). This suggested that those cells may be impacted by the disease.

When learning a scPhere model that included patient, disease status, and the anatomical region as the batch vectors, epithelial (Fig. 3b) and immune (Fig. 3c) cells grouped visually by type, with accurate cell classification (Fig. 3e, f), and the influence of region, disease status, and the patient was largely removed (Fig. 3i and Supplementary Fig. 5c, d). For example, epithelial cells were ordered in a manner consistent with their development (Fig. 3b and Supplementary Movie 3), and $CD8^+IL17^+$ T cells were nestled between $CD8^+$ T cells and activated $CD4^+$ T cells in a manner that was intriguing and consistent with the mixed features of those cells[44] (Fig. 3c).

ScPhere's abilities were particularly strong when analyzing all immune, stromal, and epithelial cells simultaneously (Fig. 3n, o, Supplementary Fig. 5e, and Supplementary Movie 4), demonstrating its capacity to embed large numbers of cells of diverse types, states, and proportions. Conversely, using $t$-SNE or UMAP with Harmony batch-corrected results of all cells as input led to an unsuccessful visualization (Fig. 3p, q): many cell subtypes from the same general compartment became indistinguishable (e.g., clumping $WNT2B^+$ fibroblasts, $RSPO3^+$ fibroblasts, and inflammatory fibroblasts), others were inexplicably split (plasma cells, which are very abundant), and yet others were adjacent even from different lineages. These results demonstrate the superior performance of scPhere compared to the combination of Harmony's batch correction and $t$-SNE or UMAP's visualization when analyzing large datasets with a large number of cells and cell types, multilevel batch effects, and complex structures (discrete cell types, continuous developmental trajectories, dominant, and rare cell subsets).

**ScPhere preserves the structure of scRNA-seq data even in very low-dimensional spaces.** We systematically assessed scPhere's performance when embedding in a latent space with few

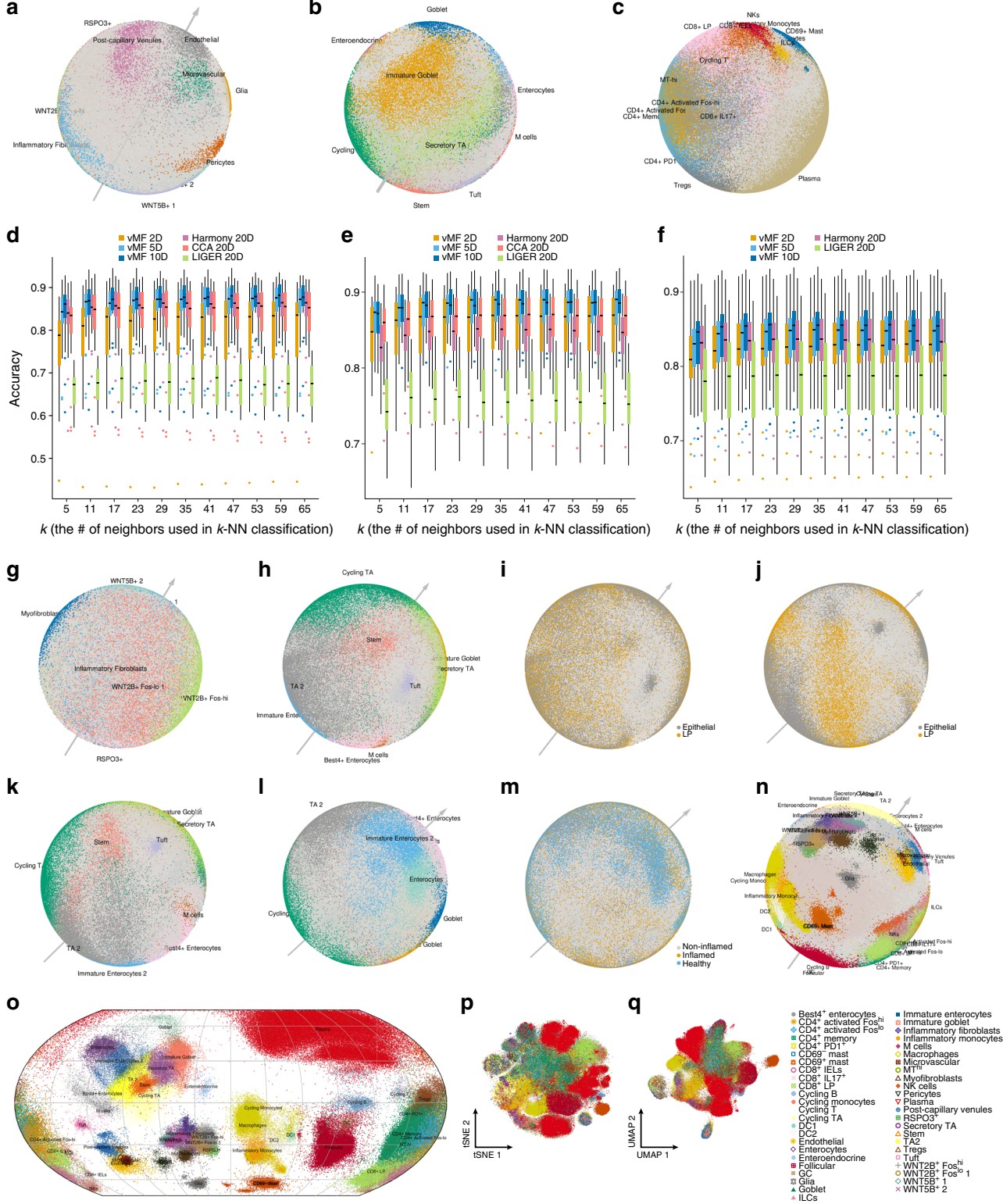

dimensions, comparing the $k$-NN classification accuracy of scPhere with hypersphere embedding to a standard normal prior and normal posteriors, which embeds cells in a Euclidean latent space, as well as to $t$-SNE, UMAP, and PHATE (holding out cells from one patient at a time for testing). We used the UC dataset, for each of the three major cell compartments separately, with the labels from the original study[33]. For $t$-SNE, UMAP, and PHATE, we used 20D or 50D Harmony batch-corrected principal components (PCs) (as Harmony can correct multilevel batch effects

and performed equally or better than LIGER[29] or Seurat3 CCA[5] on this dataset; Supplementary Figs. 6–9).

Compared to using a Euclidean latent space, when using only two dimensions (Supplementary Figs. 6–8), scPhere performed significantly better, across all $k$s (FDR < 0.05, paired $t$ test, two-tailed), suggesting that a hyperspherical latent space introduced less distortions, and is useful for data visualization. As expected, $k$-NN classification accuracies increased with the number of latent dimensions (Supplementary Figs. 6–8).

**Fig. 3 ScPhere addresses complex technical and biological batch for visualization and analysis in colon biopsies from healthy individuals and UC patients. a–c** ScPhere embedding accounting for the patient, location, and disease status. Spherical embedding of scRNA-seq profiles of (**a**) stromal, (**b**) epithelial, and (**c**) immune cells, colored by cell subset. For stromal cells, the batch vector includes only the patient. For epithelial and immune cells, patient, disease, and location are used as the batch vector. **d–f** Successful batch correction as reflected by $k$-NN classification accuracies in low dimensions. $k$-nearest neighbor classification accuracy ($y$ axis) of (**d**) stromal, (**e**) epithelial, or (**f**) immune cell types, for different $k$'s ($x$ axis), when testing on the cells from one patient, after training on the cells from all other patients, with each of several methods (color legend). Boxplots denote the medians and the interquartile ranges (IQRs). The whiskers of a boxplot are the lowest datum still within 1.5 IQR of the lower quartile and the highest datum still within 1.5 IQR of the upper quartile. **g–m** Identifying the influence of a factor on gene expression by choosing the set of batch vectors for scPhere analysis. **g** ScPhere embedding of stromal cells, with both patient and disease as the batch vector, where inflammatory fibroblasts (light blue) overlap other fibroblasts. **h–m** ScPhere embedding of epithelial cells, with either patient, disease, and location (**h**, **i**), patient and disease (**j**, **k**), or patient and location (**l**, **m**) as the batch vector. **n**, **o** Successful embedding of the entire dataset. ScPhere embedding (**n**) and its Equal Earth map projection (**o**) with all 300,000 stromal, epithelial, and immune cell profiles, colored by cell subset, with the patient, disease, and location as batch vector. **p**, **q** Limited ability of $t$-SNE and UMAP to visualize the full colon dataset. $t$-SNE (**p**) or UMAP (**q**) embedding of all 300,000 epithelial, stromal, and immune cell profiles, with Harmony batch-corrected data as input.

Overall, scPhere performed as well as $t$-SNE and UMAP based on $k$-NN accuracy or multinomial logistic regression classification accuracies, and it performed especially well for the cases with multilevel batch effects (Supplementary Figs. 7, 9b). ScPhere with hyperspherical latent spaces of dimensionality $M$ did systematically better than scPhere with Euclidean latent spaces of either dimensionality of $M$ or $M + 1$ ($M > 3$; Supplementary Fig. 7a, b). While $k$-NN accuracies increased for all methods at five latent dimensions, further increasing the latent dimensionality did not yield substantial improvements, and with further growth even decreased accuracies. Notably, even by using a 50D latent space, the $k$-NN accuracies from Harmony were worse than those from scPhere with a 5D-latent space, suggesting that scPhere captures structures in scRNA-seq data with multiple batch effects. We observed similar results for stromal (Supplementary Fig. 6) and immune (Supplementary Fig. 8) cells, and when using multinomial logistic regression instead of $k$-NN accuracies (Supplementary Fig. 9).

ScPhere's decoder that outputs a UMI count vector for each input cell can be used to impute and denoise expression values, either by sampling from the negative binormal distribution or by using the means. For example, when using the original UMI count data from $CD8^+$ T cells in the UC dataset, $CD8A$ and $CD8B$ had a Pearson correlation coefficient of only 0.27, but their decoder outputs had a Pearson correlation coefficient of 0.81 (Supplementary Fig. 11a). The $CD4$ gene, which is not expressed in $CD8^+$ T cells, was lowly expressed in both the original data and the decoder outputs, suggesting the decoder outputs did not introduce false positives.

**Querying scPhere models to recover cells impacted by different biological factors**. We next used scPhere's ability to correct for multilevel batch effects to determine which cell types were mostly influenced by specific biological factors, such as disease. We performed two analyses for this task. In the first approach, based on scPhere's ability to generate denoised outputs (Fig. 1b and Supplementary Fig. 11a), we provided both disease (healthy, uninflamed, or inflamed) and patient as the batch vector when learning a latent embedding, and obtained denoised outputs for the cells from inflamed tissues either with the original batch vector or when artificially setting "inflamed" to "healthy" in the disease batch vector (Fig. 1b). Applied to stromal and glia cells (on a 5-hypersphere), the inflammatory fibroblasts were recovered as mostly influenced by inflammation, as reflected by low correlations between the two denoised outputs (Fig. 4a). In the second approach (Fig. 4b), we trained $k$-NN classifiers using cells from both healthy and noninflamed tissue to predict cell types for cells from inflamed tissue (in the 5D hyperspherical latent space). Cell types with low true-positive rates (TPR) were likely to be the

most influenced by disease (inflammation). Indeed, inflammatory fibroblasts had very low TPRs compared to other cell types (~20%, Fig. 4c), with most misclassified as $WNT2B^+$ fibroblasts, and ~10% as $WNT5B^+$ fibroblasts (Supplementary Fig. 11b), helping assess their likely origins. The results were consistent if we only considered high-confident cells that were correctly classified when the patient was the only batch vector for scPhere analysis (Supplementary Fig. 11c).

**Batch-invariant scPhere builds atlases for annotation of unseen data**. As a parametric model, we can train scPhere to co-embed unseen (test) data to a latent space learned from training data only. To demonstrate this, we first performed a tenfold cross-validation analysis, where we partitioned the colon fibroblasts and glial cells into ten roughly equally sized subsamples, held out one subsample as out-of-sample evaluation data, and used the remaining nine subsamples as training data to select variable genes and learn different scPhere models to embed cells on a 5D hypersphere. We then trained a $k$-NN classifier on the 5D representations of the training data and used the $k$-NN classifier to classify the 5D representations of the out-of-sample evaluation data. We repeated this process ten times with each of the ten subsamples used exactly once as the out-of-sample validation data. The $k$-NN classifiers had a median accuracy of 0.834–0.853 ($k = 5$ or 65, respectively, Supplementary Fig. 11d). By comparison, when we repeat this process but using pre-computed 5D representations from all fibroblasts and glia cells, accuracy was similar (0.847–0.860, the minimal two-tailed Wilcoxon signed-rank test FDR = 0.036, and for two $k$'s, the FDRs >0.05, Supplementary Fig. 11d).

Next, we used scPhere to map cells from unseen patients, a key use case as multiple studies need to be integrated, by training a "batch-invariant" scPhere model ("Methods") that takes the gene expression vectors of cells (without batch vectors, the batch vectors were only used in the decoder part of scPhere to retain its batch-correction capabilities) as inputs and maps them to a 5D hyperspherical latent space. As a test case, we learned a batch-invariant scPhere model for stromal, epithelial, or immune cells in the 18 patients training data of the UC dataset (as in the original study[44]) and used it to map the cells from the 12 patients test data. There were multiple technical differences between the test and training data (collected nearly 2 years apart, all test cell libraries with 10× Chromium v2 chemistry, 15 of 18 training patient cell libraries with 10× Chromium v1; all test data sequenced with NextSeq but 3 of 18 training patients with HiSeq). We then trained $k$-nearest neighbor ($k$-NN) classifiers ($k = 25$) (using the labels from the original study[33]) on the 5D representations of the training data and applied the $k$-NN classifiers to the 5D representations of the test data. ScPhere's

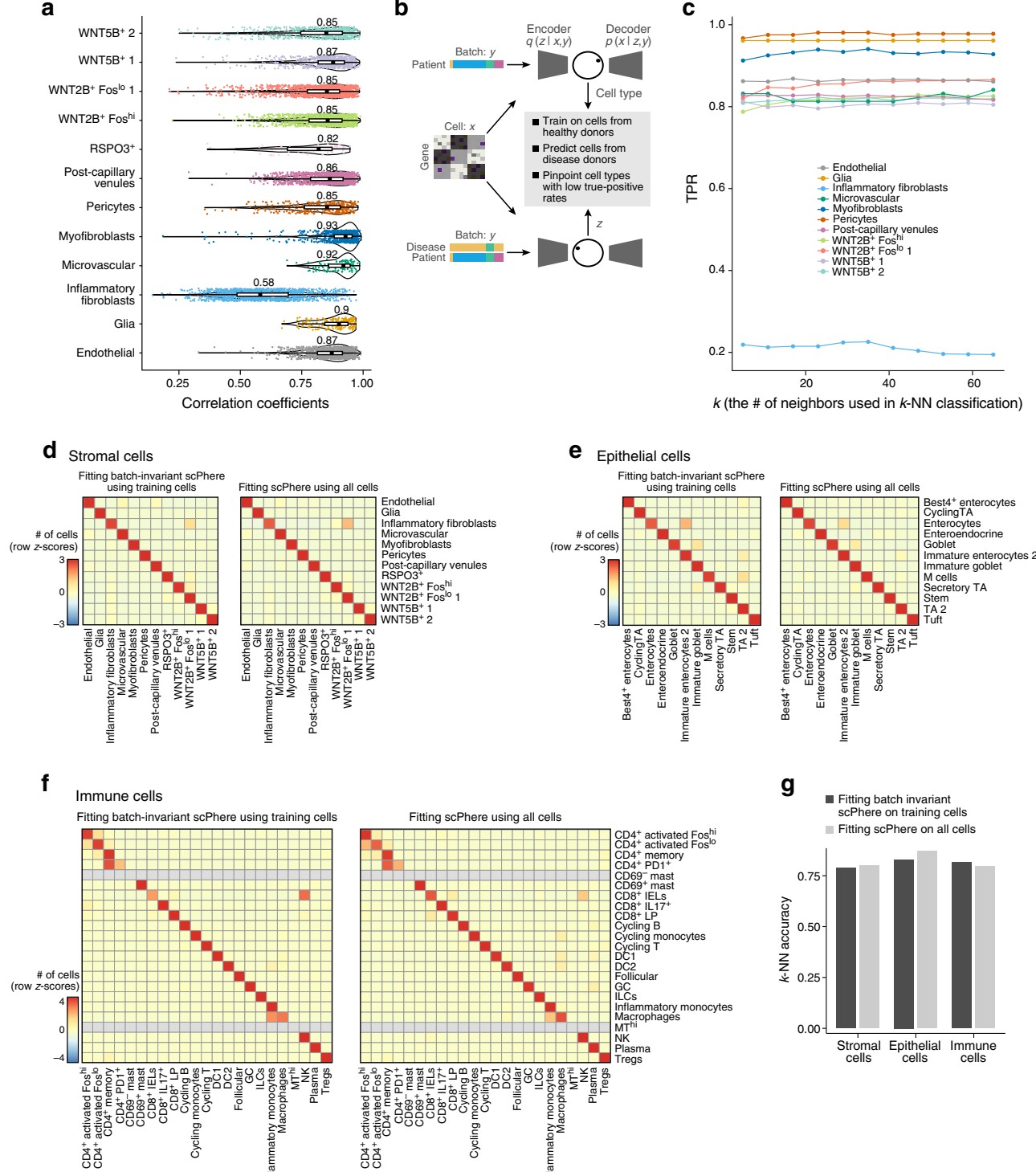

mapping of the test data was highly successful (Fig. 4d–f), with accuracies similar to those obtained when applying this process to representations from all cells (all 30 patients). Specifically, batch-invariant scPhere had accuracies of 0.79, 0.83, and 0.82 for stromal, epithelial, and immune cells, respectively, whereas a model trained on the full dataset had respective accuracies of 0.80, 0.87, and 0.80 (Fig. 4g).

**Clustering cells following scPhere embeddings**. To demonstrate how scPhere impacts clustering analysis, we clustered (using the Louvain algorithm[45,46]) the embeddings of cells on the surface of

5D hyperspheres and compared them to the clusters in the original study[44] (where only patients were used as the batch vector and variable genes were selected for each patient separately to compute a census of batch-insensitive variable genes[44]). For example, stroma and glia cells were partitioned into 18 clusters (Supplementary Fig. 12a), which were largely consistent with the original analysis[44] with some minor exceptions: *RSPO3*⁺ fibroblasts included cells from the original *WNT2B*⁺ Fos-lo cluster, and some of the inflammatory fibroblasts were in the *WNT2B*⁺ fibroblast clusters, highlighting their molecular similarity. We obtained similar results with epithelial and immune cells

**Fig. 4 Using complex batch vectors to pinpoint cell types impacted by biological factors and to generate a batch-invariant reference atlas.**
**a–c** Identification of cells impacted by disease using scPhere. **a** Imputation-based approach. Following the approach in Fig. 1b (subpanel (1)), shown is the distribution of Pearson correlation coefficients (x axis) for each cell (dot) in each type (y axis) between scPhere's denoised outputs using either the true disease batch vector or when "inflamed" was set to "healthy" in the disease batch vector. Boxplots denote the medians (labeled above) and the interquartile ranges (IQRs). The whiskers of a boxplot are the lowest datum still within 1.5 IQR of the lower quartile and the highest datum still within 1.5 IQR of the upper quartile. Violin plot width is based on a Gaussian kernel density estimation of the data. **b, c** Classification-based approach. **b** Training a classifier to infer the influence of disease on gene expression. **c** k-nearest neighbor classification true-positive rates (TPRs) (y axis) for different k's (x axis) on cells from different subsets (color legend) from inflamed tissues with a model trained using cells from healthy and noninflamed tissues. **d–g** Batch-invariant scPhere. **d–f** Confusion matrices of the overlap in cells (row-centered and scaled Z-score, color bar) between "true" cell types from the original study[44] (rows) and cell assignment by k-NN classifications (k = 25, columns) from either batch-invariant scPhere trained only on training set cells (left) or on all cells (right) for stromal (**d**), epithelial (**e**), or immune (**f**) cells. CD69⁻ mast cells and MT^hi cells were observed in the training data only. **g** The k-NN accuracies (y axis) from the confusion matrices in (**d–f**).

(Supplementary Fig. 12b, c and Supplementary Movie 5), or when we used cell embeddings on the surface of a 10D hypersphere for stromal cells (Supplementary Fig. 12d), consistent with our classification results (Fig. 3d–f). As we corrected for the influences of the region, disease, and patient, some immune cells with very similar molecular but preferentially associated with different regions (e.g., $CD69^-$ and $CD69^+$ mast cells, Supplementary Fig. 12e) or disease (cycling monocytes and macrophages, Supplementary Fig. 12f) were merged in one cluster. Notably, rare cell types were also distinct in the low-dimensional space, including cells that were missed in the original analysis (e.g., a small cluster of platelets; Supplementary Fig. 12g, cluster 33; a B-cell cluster 34 exclusively expressing $IGLC7$ and a monocyte cluster 28 expressing $FCGR3A$ and $RHOC$). UMAP, PHATE, and scPhere with normal latent variables (all in 5D) did not perform as well in some cases (Supplementary Fig. 13a–c), both by biological inspection, and by Normalized Mutual Information (NMI) and Adjusted Rand Index (ARI) (Supplementary Fig. 13d). For example, M-cells and TA2 cells were mixed in PHATE-based clustering (Supplementary Fig. 13b), and $CD8^+$ $IL17^+$ T cells and $CD4^+$ activated T cells were mixed in UMAP based clustering, as were $CD4^+$ $PD1^+$ cells and $T_{regs}$ (Supplementary Fig. 13c).

**Inferring spatial locations by embedding cells on a sphere.** The scPhere model is flexible and can be extended to additional applications, including to infer the spatial locations of cells in a tissue with the appropriate structure. To demonstrate this we focused on cells of zebrafish embryos at 50% epiboly, which are distributed on the surface of a hemisphere (or quarter sphere because of the symmetry of cell distributions), with gene expression gradients across the dorsal-ventral axis (right to left) and marginal-animal (bottom to top) axis (Fig. 5a), as well as other, punctate or salt-and-pepper patterns[47].

To map cells to a quarter sphere we forced two components of the 3D coordinates to be positive, as well as augmented the scPhere objective function to incorporate information from landmark genes (Fig. 1b, "Methods"), such that we encourage a cell expressing a given marker gene to be map within the annotated portion of the quarter sphere expressing this gene (in an 8 × 8 grid[47]). Specifically, for a cell expressing a marker gene and mapped to the quarter sphere, this modified procedure calculates the distances between its position and its annotated expression map on the grid, and minimizes the minimum of the distances. For each cell in a mini-batch training, we then calculate the sum of the minimum distances of all marker genes. The final objective function is the original scPhere objective function plus the calculated mean of the sum of minimal distances in a cell across all cells and the sum of all distances in a mini-batch training. Importantly, even if the landmarks themselves were not measured at single-cell resolution, scPhere only uses them as

weak supervision, and directly maps cells on the surface continuously, rather than to bin.

This simple modification enables scPhere to spatially map the cells successfully. We trained scPhere with only 1406 zebrafish embryonic cells[48] and 11 landmark genes[47] (Fig. 5a, "Methods"), spanning ventral, animal ventral, dorsal, animal dorsal, and marginal genes (but not animal) on an 8 × 8 grid on the quarter sphere[47]. The ventral gene marker $cdx4$, dorsal gene marker $gsc$, and marginal gene marker $osr1$ were expressed in their expected regions (Fig. 5b), as was the animal marker gene, $sox3$, even though we did not use any animal genes in the training (Fig. 5b). After mapping cells to the quarter sphere, we next calculated the spatial gene expression patterns[47], and the results (Fig. 5c) matched the expected patterns (Fig. 5a). We then used the trained scPhere model to map 3820 cells from another three batches (Fig. 5d), obtaining consistent spatial patterns (Fig. 5b). Finally, from the mapped cells, we could correctly predict patterns for genes not included in the training, including salt-and-pepper patterns and random sparse patterns for "apoptotic-like" cells (Supplementary Fig. 14). Notably, this mapping approach can be extended to nonspherical shapes by transforming the cells distributed on a plane to complex shapes (see "Discussion").

**Embedding cells in a hyperbolic space for trajectory discovery and interpretation.** When cells are expected to show developmental trajectories, such as from adult stem cells to differentiated cells, scPhere can embed them into a hyperbolic space of the Lorentz model[24,25], and optionally convert the coordinates in the Lorentz model to the Poincaré disk for 2D visualization[34,49]. Moreover, if we position the expected root cells of the developmental process at the center of a Poincaré disk, then the distance of each cell from the center can be thought of as a pseudotime[20,26,50]. For a specific cell type, we can see cells progress with distance and angles continuously in the Poincaré disk. We can also encourage mapping root cells (if they are known a priori) to the origin of the Lorentz model during training.

Applying this first to colon epithelial cells, we readily discerned developmental ordering from intestinal stem cells to terminally differentiated cells in either the Poincaré disk (Fig. 6a), with stem cells at the center of the disk for intuitive interpretation, or in the Lorentz model (Supplementary Fig. 15a): the two major cell development trajectories are clearly delineated (Fig. 6a, arrows connecting median coordinates of cells of different types) and M-cells and Best$^+$ enterocytes are close to each other. PHATE[37] visualization using the 5D representations of cells in the Lorentz model as inputs recapitulated the results from the 2D representations (Supplementary Fig. 15b). In contrast, developmental trajectories were less apparent when we embedded cells in Euclidean space (Fig. 6b) or when we used PHATE multidimensional scaling on the 5D representations of cells in the Euclidean space (Supplementary Fig. 15c), with cells in the two

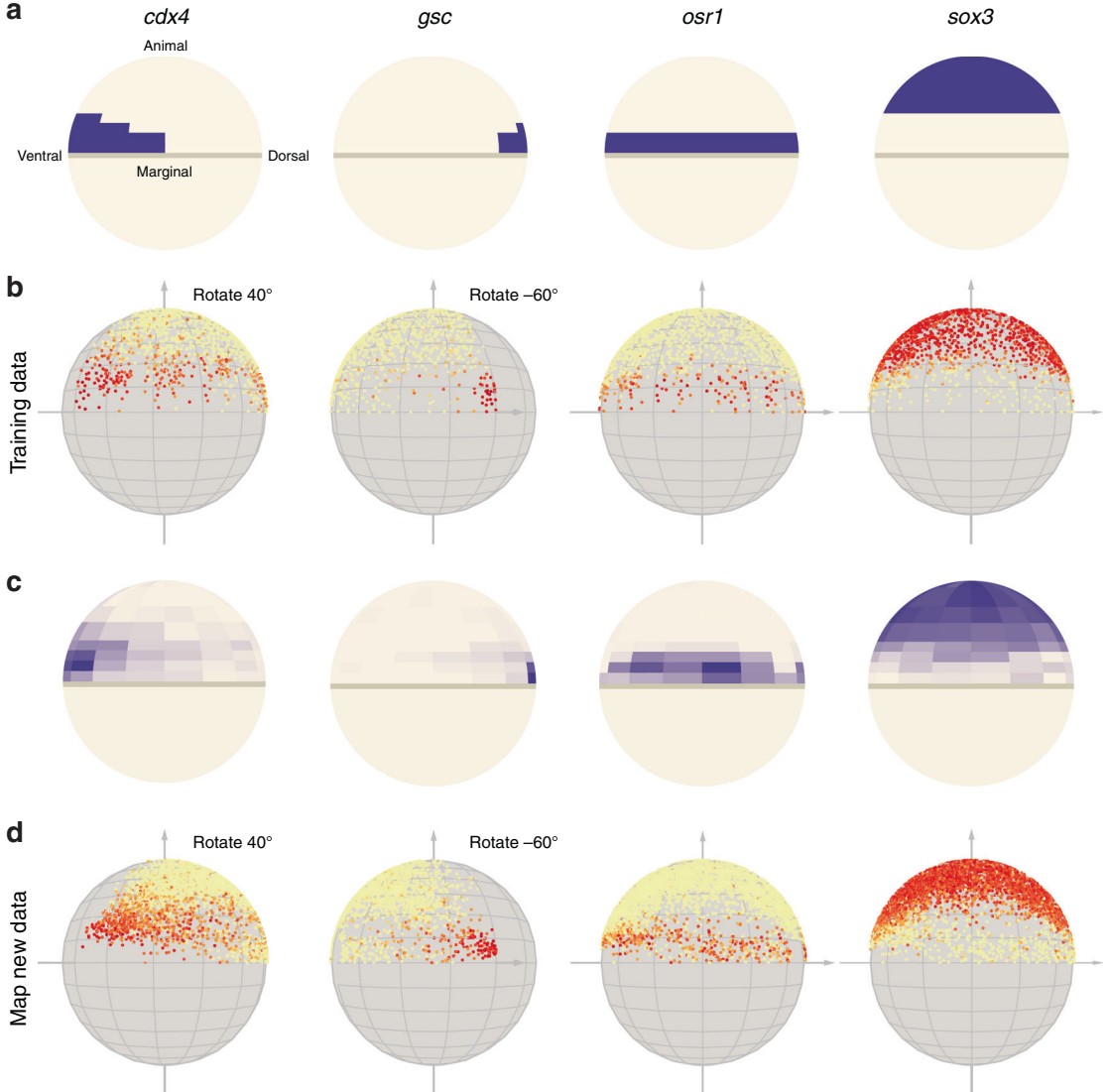

**Fig. 5 Spatial mapping with scPhere.** Representative marker gene expression from zebrafish in 50% epiboly on quarter spheres (**a**, **c**) and scPhere inferred spatial locations colored by marker gene expression (**b**, **d**) for the original marker expression (**a**) inferred expression after mapping cells to quarter spheres (**c**), and for 1406 cells in the training set (**b**) or another 3820 cells from three test batches (**d**).

major developmental branches being close to each other. 2D visualization with *t*-SNE, UMAP, and PHATE was reasonable (Supplementary Fig. 15d–f), although the *t*-SNE had some small spurious clusters, in the UMAP cycling TAs were intermediate between stem cells and secretory TAs (which can differentiate directly), and in PHATE several cell types were merged (M-cells and TA2 cells, tuft and enteroendocrine cells).

Next, we analyzed 86,024 *C. elegans* embryonic cells[51] collected along a time course from <100 min to >650 min after each embryo's first cleavage, finding that cells were ordered neatly in the latent space by both time and lineage, from a clearly discernible root at time 100–130 at the center of the Poincaré disk (cells from <100 were mostly unfertilized germline cells, "Methods") to cells from time >650 near the border of the Poincaré disk (Fig. 6c, d and Supplementary Fig. 16) or away from the origin in the Lorentz model (Supplementary Fig. 17a, b). Within the same cell type, cells were ordered by embryo time in the Poincaré disk (Fig. 6d) or in the Lorentz model (Supplementary Fig. 17a, b). After first appearing along a developmental trajectory, cells of the same type progressed with embryo time, forming a continuous trajectory occupying a range of angles. For

example, the cells of the body wall muscle (BWM, the most abundant cell type in this dataset, Supplementary Fig. 16) first appeared at embryo time 130–170 in a separable position (bottom left of the Poincaré disk, Fig. 6e), and then "advance" toward bottom right of the Poincaré disk in a continuous progression but in a manner aligned with embryo time (i.e., from 170–210 to >650) and lineages (i.e., from first row and second row BWMs (MS lineage) to anterior (MS to D lineage), and to posterior BWMs (C lineage)[51], Supplementary Fig. 17c). Moreover, different cell types (e.g., ciliated amphid neurons, ciliated nonamphid neurons, hypodermis, G2 and W blasts, seam cells, body wall muscle) that appeared at slightly different embryonic time points, had their origins around the same region and progressed with embryonic time in a similar way, forming a continuous trajectory but at a different angle and/or distance ranges from the center (Fig. 6d, arrows). Accordingly, cells' distances to the origin were correlated with their embryonic time (Pearson correlation coefficients = 0.55, Supplementary Fig. 17d). For a few rare cell types that appeared relatively late in a developmental trajectory, such as coelomocytes (appearing in 270–330), their distances to the origin could be negatively

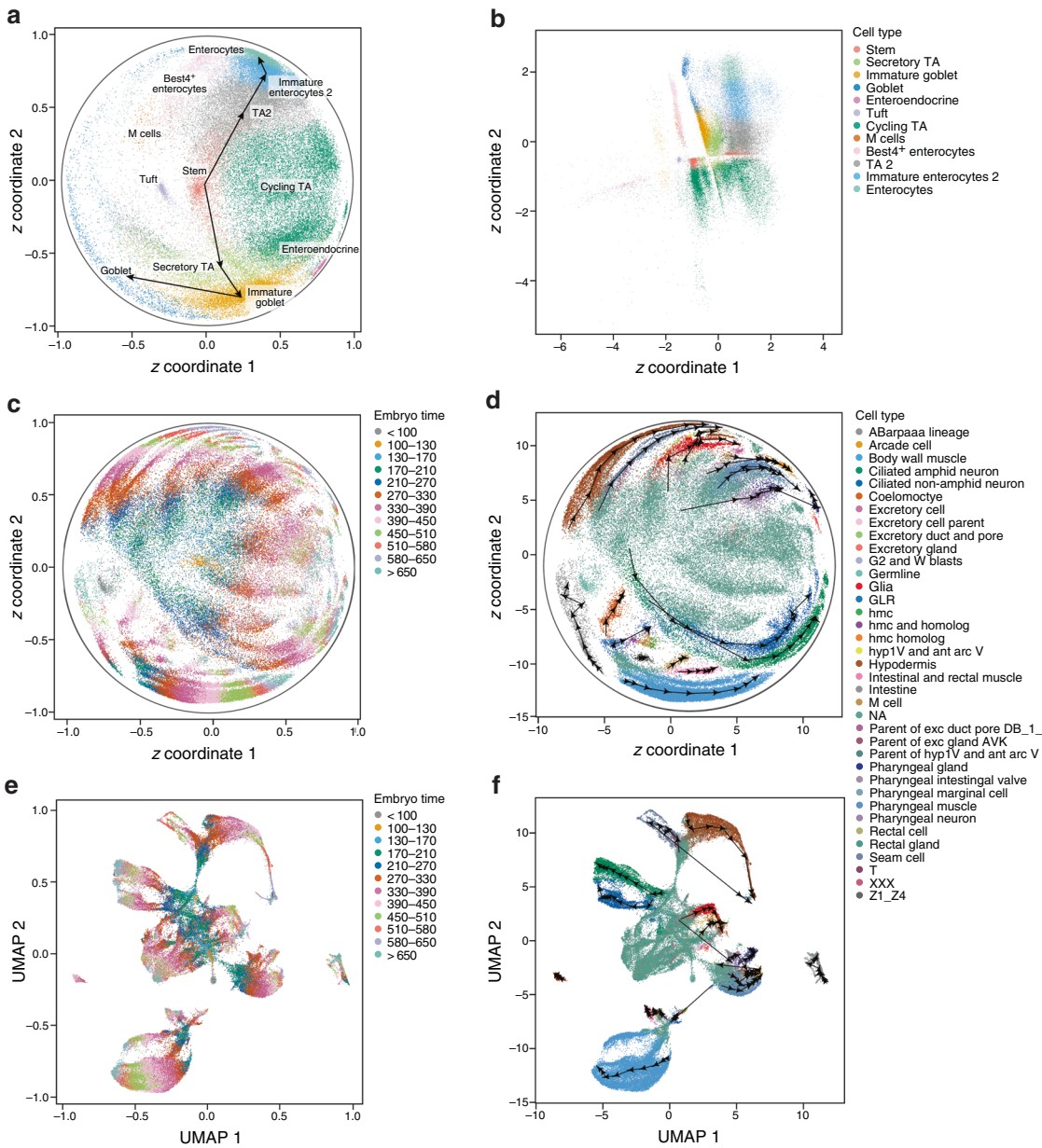

**Fig. 6 Embedding cells in the hyperbolic space for trajectory discovery and interpretation. a**, **b** Major development trajectories of colon epithelial cells are discerned from embeddings in the Poincaré disk. Single-epithelial cell profiles (dots) annotated by type as embedded in a Poincaré disk (**a**) or the Euclidean space (**b**). Arrows point to the next cell type in the development trajectory. **c**–**f** Embedding of cells from a *C. elegans* embryonic time course highlights developmental timing and differentiation to subsets. Single-cell profiles (dots) annotated by either time (**c**, **e**) or cell type (**d**, **f**) in a hyperbolic space of the Poincaré disk (**c**, **d**) or in a 2D UMAP (with 50 principal components, batch-corrected by Harmony) (**e**, **f**). Arrows connect the median coordinates of cells of a type in two consecutive time points.

correlated with embryonic times, and re-centering their embeddings can help interpret their trajectories "locally".

These patterns are harder to discern in UMAP, *t*-SNE, or PHATE (Fig. 6e, f, with 50 batch-corrected PCs by Harmony as inputs; Supplementary Figs. 18a, b and 19), where cells from consecutive time points were compacted, cells that appeared early were relatively distant from each other in the embeddings, and temporal progression was not in the same direction. Moreover, when we quantify time continuity, by comparing the *k*-nearest neighbor time point classification accuracies (in a tenfold cross-validation analysis), accuracies from scPhere (in 2D) were higher than those from *t*-SNE, UMAP, and PHATE (in 2D, Supplementary Fig. 18c). Thus, a scPhere model with a hyperbolic latent space learned smooth (in time) and interpretable cell trajectories

and helped represent developmental and other temporal processes.

## Discussion

We introduced scPhere, a deep-generative model to embed single cells on hyperspheres or in hyperbolic spaces to enhance exploratory data analysis and visualization of cells from single-cell studies, especially with complex, multilevel batch factors. ScPhere provides more readily interpretable representations, and avoids occlusion, as we demonstrate in diverse systems, and, when embedding cells in hyperbolic spaces, it helps studying developmental trajectories. In this latter case, in addition to providing compelling visualizations compared to state-of-the-art methods, by placing root cells at the center of a Poincaré disk, we derived a

natural definition for pseudotime as the distance to the center. The cells of type progress continuously with distance and angle in the Poincaré disk.

A major advantage of scPhere is in effectively accounting for multilevel complex batch effects, which we show disentangles cell types from patients, diseases, and location variables. We can harness this ability in several ways: to visualize and cluster cells while controlling for one or more factors, and examining the influence of any combination of them; to investigate which cell types are most affected by a factor (e.g., disease status, or location); or to generate batch-invariant reference embeddings, to which additional data can be mapped from new individuals, samples or conditions. In this study, we parameterize the dispersion parameters of the negative-binomial distributions as functions of cell count vectors. We may let the dispersion parameters as fixed values and optimize them directly for some tasks (not functions of cell count vectors). ScPhere's ability to handle complex batch factors is an advantage over previous methods for batch correction (e.g., SAUCIE[27], scVI[12], LIGER[29], Seurat3 CCA[5], fastMNN[19], Scanorama[28], and Conos[31]), which handle only one batch vector. Indeed, in our benchmarking with IBD cells with 30 patients, three disease statuses, and two spatial locations, scPhere performed better than state-of-the-art batch-correction methods such as Harmony, Seurat3 CCA, LIGER. In the future, we can leverage supervised information to further estimate the uncertainty of aligning cells from batches. In addition, as a parametric model, scPhere can naturally co-embed unseen (test) data to a latent space learned from training data only, and denoise expression data successfully.

ScPhere is especially useful for analyzing large scRNA-seq datasets: It is efficient, as it scales linearly with the number of input cells; it does not suffer from "cell-crowding" even with large numbers of input cells; and it better preserves hierarchical, global structures in data than competing methods. Finally, by learning a "batch-invariant" encoder that takes gene expression as inputs to learn latent embeddings, it forms a reference to annotate new profiled cells from future studies[52]. This is another major advantage over nonparametric methods such as t-SNE, UMAP, and Poincaré maps, which do not have a natural way to embed new data, especially in the presence of batch effects, and have scalability issue. These features should make it well-suited for the challenge of building a comprehensive reference map, in health studies, such as the Human Cell Atlas[3], as well as in diseases, such as in the Human Tumor Atlas Network[53].

The scPhere model is robust to hyperparameters. Here, we used the same hyperparameters for scPhere analyses for all nine datasets (varying from ~1000 to >300,000 cells), whereas some previous studies[54] showed that classical variational autoencoders could be sensitive to hyperparameters. ScPhere's robustness may stem from the robust negative-binomial distribution for modeling UMI counts, or from the use of non-Euclidean latent spaces to help solve the cell-crowding problem in the latent space.

One key extension we have shown for scPhere is modifying it to spatially map cells. As our first example, we mapped zebrafish embryonic cells to a quarter sphere to infer the spatial locations of cells in a tissue, because a sphere is an appropriate model at this developmental phase. The only extra input we provided was the (binned) spatial expression patterns of a handful of landmark genes[47]. The resulting model retains scPhere's scalability and parametric nature, which allows mapping new cells. Importantly, this approach can be readily extended to other tissues with complex, nonspherical shapes (e.g., the mouse hippocampus), by transforming the cells distributed on a plane to such complex shapes, using methods such as normalizing flow[55]. Our approach to spatial mapping is distinct in that we use the global shape of the physical space as a constraint, whereas most approaches do not consider this at all, and those that do, like novoSPARC[56], only incorporate continuity assumptions, which cannot capture many spatial patterns.

ScPhere can be extended in several other ways. When cell type annotations or cell-type marker genes for some of the analyzed cells are available, we can include semi-supervised learning to annotate cell types[57,58]. Although scPhere showed promising denoising results, further studies are required to explore its abilities in imputing missing counts in scRNA-seq data and removing ambient RNA contamination[59,60]. Given the rapid development of spatial transcriptomics[61,62], single-cell ATAC-seq[63,64], and other complementary measurements, scPhere can be extended for integrative analysis of multimodal data. We can also learn discrete hierarchical trees for better interpreting developmental trajectories, use more complex topological latent spaces such as tori with diffuse VAEs[65], and even learn optimal latent spaces using mixture curvature VAEs[66]. Additional developments can extend scPhere to model perturbation data. Moreover, there are not yet many tools to process data distributed in the hyperbolic space, such as efficient $k$-NN search tools, and future studies can address this gap. Given its scope, flexibility, and extensibility, we foresee that scPhere will be a valuable tool for large-scale single-cell and spatial genomics studies.

## Methods

**Mapping scRNA-seq data to a hyperspherical latent space**. ScPhere received as input a scRNA-seq dataset $D = \{(\mathbf{x}_i, \mathbf{y}_i)_{i=1}^N\}$, where $\mathbf{x}_i \in \mathbb{R}^D$ is the gene expression vector of cell $i$, $D$ is the number of measured genes, $\mathbf{y}_i$ is a categorical variable vector specifying the batch in which $\mathbf{x}_i$ is measured, and $N$ is the number of cells. Although $\mathbf{x}_i$ is high-dimensional, its intrinsic dimensionality is typically much lower. We therefore assume that the $\mathbf{x}_i$ distribution is governed by a much lower-dimensional vector $\mathbf{z}_i$, and the joint distribution is factorized as follows (Fig. 1a):

$$p(\mathbf{x}_i, \mathbf{y}_i, \mathbf{z}_i | \boldsymbol{\theta}_i) = p(\mathbf{y}_i | \boldsymbol{\theta}_i) p(\mathbf{z}_i | \boldsymbol{\theta}_i) p(\mathbf{x}_i | \mathbf{y}_i, \mathbf{z}_i, \boldsymbol{\theta}_i)$$

Here $p(\mathbf{y}_i | \boldsymbol{\theta}_i)$ is the categorical distribution, $p(\mathbf{z}_i | \boldsymbol{\theta}_i)$ is the prior distribution for $\mathbf{z}_i$ ($\mathbf{z}_i \in \mathbb{R}^M$, $\mathbf{z}^T\mathbf{z} = 1$, $M \ll D$), which is assumed to be a uniform distribution on a hypersphere with density $\left(\frac{2(\pi^{M/2})}{\Gamma(M/2)}\right)^{-1}$. For notational simplicity, we use bold font $\boldsymbol{\theta}_i$ to represent the parameters of each distribution, e.g., the parameters $\boldsymbol{\theta}_i$ in $p(\mathbf{y}_i | \boldsymbol{\theta}_i)$ and $p(\mathbf{z}_i | \boldsymbol{\theta}_i)$ are the parameters of the two distributions and should be different.

For scRNA-seq data, the observed Unique Molecular Identifier (UMI) count of gene $j$ in cell $i$ has typically been assumed to follow a zero-inflated negative-binomial (ZINB) distribution[11,12,67]. However, a recent study suggests that zero inflation is an artifact of normalizing UMI counts[68], and negative-binomial distributions generally fit the UMI counts well[69–71]. We, therefore, assume a negative-binomial distribution of observations in this study:

$$p(\mathbf{x}_i | \mathbf{y}_i, \mathbf{z}_i, \boldsymbol{\theta}_i) = \prod_{j=1}^D \text{NB}(x_{i,j} | \mu_{\mathbf{y}_i, \mathbf{z}_i}, \sigma_{\mathbf{y}_i, \mathbf{z}_i})$$

The negative-binomial parameters mean $\mu_{\mathbf{y}_i, \mathbf{z}_i} > 0$ and dispersion $\sigma_{\mathbf{y}_i, \mathbf{z}_i} > 0$ are specified by a model neural network (decoder), which can model complex nonlinear relationships between the latent variables and the observations.

We next want to compute the posterior distribution $p(\mathbf{z}_i | \mathbf{y}_i, \mathbf{x}_i, \boldsymbol{\theta}_i)$, which is assumed to be a von Mises–Fisher (vMF) distribution on a unit hypersphere of dimensionality $M - 1$: $\mathbb{S}^{M-1} = \{\mathbf{z} | \mathbf{z} \in \mathbb{R}^M, \mathbf{z}^T\mathbf{z} = 1\}$. We turn to variational inference to find a $q(\mathbf{z}_i | \mathbf{y}_i, \mathbf{x}_i, \boldsymbol{\phi}_i)$ to approximate the posterior, since exact inference is intractable, given that the model is parameterized by a neural network. In addition, the number of parameters to estimate grows with the number of cells, because each cell has a "local" distribution with parameter $\boldsymbol{\phi}_i$. To scale to large datasets, variational autoencoders use an inference neural network (encoder, with a fixed number of parameters) to output the "local" parameter $\boldsymbol{\phi}_i$ of each cell. Therefore, the learning objective is to find the model neural network and the inference neural network parameters to maximize the evidence lower bounds:

$$\mathcal{L}(\boldsymbol{\theta}_i, \boldsymbol{\phi}_i) = -\mathbb{KL}\big(q(\mathbf{z}_i | \mathbf{y}_i, \mathbf{x}_i, \boldsymbol{\phi}_i) || p(\mathbf{z}_i | \boldsymbol{\theta}_i)\big) + \mathbb{E}_{q(\mathbf{z}_i | \mathbf{y}_i, \mathbf{x}_i, \boldsymbol{\phi}_i)}[p(\mathbf{x}_i | \mathbf{y}_i, \mathbf{z}_i, \boldsymbol{\theta}_i)] \quad (1)$$

The Kullback–Leibler ($\mathbb{KL}$) divergence[72] in Eq. (1) can be calculated analytically (below). We use Monte–Carlo integration (sampling from the vMF distribution $q(\mathbf{z}_i | \mathbf{y}_i, \mathbf{x}_i, \boldsymbol{\phi}_i)$) to calculate the second term.

To make scPhere robust to small perturbations (e.g., sequencing depth) and to stabilize training, we add a penalty term to the objective function in Eq. (1). Specifically, for each gene expression vector $\mathbf{x}_i$, we downsample $\mathbf{x}_i$ by keeping 80% (downsampling ratio 20%) of its UMIs to produce $\mathbf{x}_i$. The latent representations of $\mathbf{x}_i$

and $\hat{\mathbf{x}}_i$ are $\mathbf{z}_i$ and $\hat{\mathbf{z}}_i$, respectively. The penalty term is defined as $-\sum_{j=1}^{M}(z_{i,j}-\hat{z}_{i,j})^2$ as we want $\mathbf{z}_i$ and $\hat{\mathbf{z}}_i$ to be close. Even when increasing the downsampling ratio to 50 or 80%, scPhere (with hyperspherical latent spaces, such that the distance between two points is less than or equal to 2) produced similar results as reflected by k-NN accuracies (Supplementary Fig. 20a). K-NN accuracies were the lowest when removing this penalty term (i.e., downsampling ratio = 0). For hyperbolic latent spaces, adding this term helps stabilize training. Otherwise, the ELBO is more likely to become NA during training. For example, for the mouse retina neurons dataset, the ELBO became NA after training ~80,000 mini-batches without this penalty term.

The von Mises–Fisher (vMF)[73] distribution represents angular observations as points on the surface of a unit-radius hypersphere. Let $\mathbf{z}$ be a $M$-dimensional random vector with unit radius ($\mathbf{z}^T\mathbf{z}=1$), then its probability density function is:

$$\mathrm{vMF}(\mathbf{z}|\boldsymbol{\mu},\kappa)=C_m(\kappa)\exp(\kappa\boldsymbol{\mu}^T\mathbf{z})$$
$$C_m(\kappa)=\kappa^{M/2-1}(2\pi)^{-M/2}I_{M/2-1}^{-1}(\kappa)$$

where $\boldsymbol{\mu}^T\boldsymbol{\mu}=1$ is the mean direction vector (not the mean) and $\kappa\geq 0$ is the concentration parameter. The greater the value of $\kappa$, the higher the concentration of distribution around the mean direction vector $\boldsymbol{\mu}$. When $\kappa=0$, $\mathrm{vMF}(\mathbf{z}|\boldsymbol{\mu},0)=\left(\frac{2(\pi^{M/2})}{\Gamma(M/2)}\right)^{-1}$ is the uniform distribution on the unit hypersphere $\mathbb{S}^{M-1}$. $C_m(\kappa)$ is a constant normalization factor and $I_\nu(\cdot)$ is the modified Bessel function of the first kind of order $\nu$[74]: $I_\nu(\kappa)=(\frac{\kappa}{2})^\nu\sum_{t=0}^{\infty}\frac{(\kappa^2/4)^t}{t!\Gamma(\nu+t+1)}$. The Gamma function is defined as $\Gamma(x)=\int_0^\infty s^{x-1}e^{-s}ds$.

For random vectors distributed on the surface of a hypersphere, a natural prior is the uniform distribution, which is the vMF distribution with zero concentration $\mathrm{vMF}(\mathbf{z}|\boldsymbol{\mu},0)$. In this case, the Kullback–Leibler ($\mathbb{KL}$) divergence[52] can be written in closed-form:

$$\mathbb{KL}(\mathrm{vMF}(\mathbf{z}|\boldsymbol{\mu},\kappa)||\mathrm{vMF}(\mathbf{z}|\boldsymbol{\mu},0))=\int_{\mathbb{S}^{M-1}}\mathrm{vMF}(\mathbf{z}|\boldsymbol{\mu},\kappa)\log\frac{\mathrm{vMF}(\mathbf{z}|\boldsymbol{\mu},\kappa)}{\mathrm{vMF}(\mathbf{z}|\boldsymbol{\mu},0)}d\mathbf{z}$$
$$=\int_{\mathbb{S}^{M-1}}\mathrm{vMF}(\mathbf{z}|\boldsymbol{\mu},\kappa)(\log C_M(\kappa)+\kappa\boldsymbol{\mu}^T\mathbf{z}+\log 2+\log(\pi)M/2-\log\Gamma(M/2))d\mathbf{z}$$
$$=\log C_M(\kappa)+\kappa\boldsymbol{\mu}^T\boldsymbol{\mu}\frac{I_{M/2}(\kappa)}{I_{M/2-1}(\kappa)}+\log 2+\log(\pi)M/2-\log\Gamma(M/2)$$

(2)

with

$$\log C_M(\kappa)=(M/2-1)\log(\kappa)-(M/2)\log(2\pi)-\log(I_{M/2-1}(\kappa))$$

Notice that Eq. (2) is independent of the mean direction vector $\boldsymbol{\mu}$ as $\boldsymbol{\mu}^T\boldsymbol{\mu}=1$, so we only need to take the derivative of Eq. (2) w.r.t $\kappa$ during optimization. In other words, minimizing the $\mathbb{KL}$ divergence only forces the concentration parameter $\kappa$ to be close to zero but without any forces on the mean direction vector. This is different from using a location-scale family of priors, such as a standard normal prior, where the prior encourages the posterior means of all points to be close to zero. When $\nu\ll\kappa$, $I_\nu(\kappa)$ overflows quite rapidly with $\kappa$. To avoid numeric overflow, we use the exponentially scaled modified Bessel function $e^{-\kappa}I_\nu(\kappa)$ in calculations (the scaling is motivated by the asymptotic expansion of $I_\nu(\kappa)\sim e^\kappa(2\pi\kappa)^{-1/2}\sum_t\alpha_t(\nu)\kappa^{-t}$ for $\kappa\to\infty$[75]). The first-order derivative of the exponentially scaled modified Bessel function is

$$\frac{de^{-\kappa}I_\nu(\kappa)}{d\kappa}=e^{-\kappa}\left(-I_\nu(\kappa)+I_{\nu-1}(\kappa)-\frac{\nu}{\kappa}I_\nu(\kappa)\right)$$

Previous work has used vMF distribution as the latent distribution for variational autoencoders[25,32,33], but only the spherical variational auto-encoder[23] learns the concentration parameter $\kappa$.

Samples from vMF distributions can be obtained through a rejection sampling scheme[76,77]. The algorithm is based on the theorem[77] that a $M$-dimensional vector $\mathbf{z}=(\sqrt{1-\omega^2}\mathbf{v}^T,\omega)^T$ has a vMF distribution with direction vector $(0,\ldots,1)^T\in\mathbb{S}^{M-1}$ and concentration parameter $\kappa$ if $\omega$ has a univariate density function with the following density function:

$$f(\omega)=\frac{e^{\kappa\omega}(1-\omega^2)^{(M-3)/2}}{C_\kappa B(\frac{1}{2},\frac{1}{2}(M-1))},\omega\in(-1,1),M\geq 2$$ (3)

where $\mathbf{v}$ is uniformly distributed in $\mathbb{S}^{M-2}$, $C_\kappa$ is a normalization term such that $f(\omega)$ is a legitimate density function, and $B(x,y)=\frac{\Gamma(x)\Gamma(y)}{\Gamma(x+y)}$ is the Beta function. The vector $\mathbf{v}$ is uniformly distributed in $\mathbb{S}^{M-2}$ and can be sampled from a standard normal distribution in $M-1$ dimensions and then we normalize the resulting sample to unit length.

We then use rejection sampling to sample $\omega$ from the univariate distribution in Eq. (3). The envelope function used for rejection sampling is defined as

$$g(\omega)=\frac{2b^{(M-1)/2}}{B(\frac{1}{2}(M-1),\frac{1}{2}(M-1))}\frac{(1-\omega^2)^{(M-3)/2}}{((1+b)-(1-b)\omega)^{M-1}},\quad\omega\in(-1,1),M\geq 2$$

(4)

Where the term[78] $b=\frac{M-1}{2\kappa+\sqrt{4\kappa^2+(M-1)^2}}$. To sample from $g(\omega)$, we can first sample $\varepsilon\sim\mathrm{Beta}(\frac{M-1}{2},\frac{M-1}{2})$ and pass the sample $\varepsilon$ to the invertible function $h(\varepsilon)=\frac{1-(1-b)\varepsilon}{1-(1+b)\varepsilon}$. We can easily prove that $\omega=\frac{1-(1-b)\varepsilon}{1-(1+b)\varepsilon}$ is distributed according to Eq. (4) based on the rule of transforming a continuous random variable with an invertible function. A sample $\omega$ is accepted if $\kappa\omega+(M-1)\log(1-x_0\omega)-c\geq\log(u)$, where $x_0=\frac{1-b}{1+b}$, and $c=\kappa x_0+(M-1)\log(1-x_0^2)$ and $u$ is sampled from a continuous uniform distribution with support in $(0,1)$. The vector $\mathbf{z}'=(\sqrt{1-\omega^2}\mathbf{v}^T,\omega)^T$ is a sample from $\mathrm{vMF}(\mathbf{z}'|\mathbf{e}_1,\kappa)$, where $\mathbf{e}_1=(0,\ldots,1)^T\in\mathbb{S}^{M-1}$. We can then rotate $\mathbf{z}'$ using a Householder matrix $\mathbf{I}-\mathbf{u}\mathbf{u}^T$ to get a sample from $\mathrm{vMF}((\mathbf{z}|\boldsymbol{\mu},\kappa)$[23], where $\mathbf{I}$ is the identify matrix of rank $M$ and $\mathbf{u}=\frac{\mathbf{e}_1-\boldsymbol{\mu}}{||\mathbf{e}_1-\boldsymbol{\mu}||}$, where $||\cdot||$ is the Euclidean norm. Overall, the samples from a Beta distribution are transformed and accepted or rejected by the rejection sampling scheme, and combined with samples $\mathbf{v}$ from a uniform distribution in $\mathbb{S}^{M-2}$. The combined samples are further transformed to generate samples from the desired vMF distribution. Remarkably, previous work has shown that this reparameterization approach still holds for these samples[23] and can be used to optimize the vMF parameters $\boldsymbol{\mu}$ and $\kappa$, which are the outputs of the inference neural network (encoder).

For visualization purposes, we typically set $M=3$. Then the univariate density function becomes $f(\omega)=\frac{e^{\kappa\omega}}{C_\kappa B(\frac{1}{2},1)}=\frac{\kappa}{e^\kappa-e^{-\kappa}}e^{\kappa\omega}=\frac{\kappa}{2\sinh(\kappa)}e^{\kappa\omega}$, where $\sinh(\cdot)$ is the hyperbolic sine function. We can directly draw samples from this density function by transforming a sample $\xi$, generated from a continuous uniform distribution $\xi\sim\mathrm{Unif}(0,1)$ using the inverse cumulative function $F(t)=\int_{\omega=-1}^{t}\frac{\kappa}{2\sinh(\kappa)}e^{\kappa\omega}d\omega=\frac{1}{2\sinh(\kappa)}(e^{\kappa t}-e^{-\kappa})$. Specifically, we can use the following algorithm to generate a sample from $f(\omega)$:

$$\xi\sim\mathrm{Unif}(0,1)\,\omega=F^{-1}(\xi)=1+\frac{\log(\xi-\xi e^{-2\kappa}+e^{-2\kappa})}{\kappa}$$

**Poincaré ball and Lorentz model of the hyperbolic space.** The Poincaré ball model represents the hyperbolic space as the interior of a unit ball in the Euclidean space: $\mathbb{P}=\mathbf{z}\in\mathbb{R}^{M+1}|\,\|\mathbf{z}\|<1, z_0=0, M\in\mathbb{Z}^+$, where $\mathbf{z}=(z_0,\ldots,z_M)^T$. The distance between two points $\mathbf{z}_1,\mathbf{z}_2\in\mathbb{P}$ is defined as:

$$d_\mathbb{P}(\mathbf{z}_1,\mathbf{z}_2)=\cosh^{-1}\left(1+\frac{2\|\mathbf{z}_1-\mathbf{z}_2\|^2}{(1-\|\mathbf{z}_1\|^2)(1-\|\mathbf{z}_2\|^2)}\right)$$

where $\cosh^{-1}(z)=\ln(z+\sqrt{z^2-1})$ is the inverse hyperbolic cosine function, which is monotonically increasing for $z\geq 1$. The symbol $\|\cdot\|$ represents the Euclidean norm. Notice that $\cosh^{-1}(1+z)=\ln(1+z+\sqrt{z^2+2z})$, which approximates $\sqrt{2z}$ when $\lim z\to 0$ and $\ln(2z)$ for $\lim z\to +\infty$. When both $\mathbf{z}_1$ and $\mathbf{z}_2$ are close to the origin with zero norm, $d(\mathbf{z}_1,\mathbf{z}_2)\approx\cosh^{-1}(1+2\|\mathbf{z}_1-\mathbf{z}_2\|^2)\approx 2\|\mathbf{z}_1-\mathbf{z}_2\|$. Therefore, the Poincaré ball model resembles Euclidean geometry near the center of the unit hyperball. The induced norm of a point $\mathbf{z}\in\mathbb{P}$ is

$$\|\mathbf{z}\|_\mathbb{P}=\cosh^{-1}\left(\frac{1+\|\mathbf{z}\|^2}{1-\|\mathbf{z}\|^2}\right)$$

As $\mathbf{z}$ moves away from the origin and approaches the border with $\|\mathbf{z}\|\approx 1$, the induced norm $\|\mathbf{z}\|_\mathbb{P}$ grows exponentially. Hyperbolic geometry is useful to represent data with an underlying approximate hierarchical structure.

The Lorentz model is a model of the hyperbolic space and points of this model satisfy $\mathbb{H}^M=\{\mathbf{z}\in\mathbb{R}^{M+1}|z_0>0,\langle\mathbf{z},\mathbf{z}\rangle_\mathbb{H}=-1\}$, where $\langle\mathbf{z},\mathbf{z}'\rangle_\mathbb{H}=-z_0z_0'+\sum_{i=1}^{M}z_iz_i'$ is the Lorentzian inner product (or Minkowski inner product when $\mathbf{z}\in\mathbb{R}^4$). The special one-hot vector $\boldsymbol{\mu}_0=(1,0,\ldots,0)^T$ is the origin of the hyperbolic space. The distance between two points of the Lorentz model is defined as:

$$d_\mathbb{H}(\mathbf{z}_1,\mathbf{z}_2)=\cosh^{-1}\left(-\langle\mathbf{z}_1,\mathbf{z}_2\rangle_\mathbb{H}\right)$$

The tangent space of $\mathbb{H}^M$ at point $\boldsymbol{\mu}\in\mathbb{H}^M$ is defined as $\mathcal{T}_{\boldsymbol{\mu}}\mathbb{H}^M:=\{\mathbf{z}|\langle\boldsymbol{\mu},\mathbf{z}\rangle_\mathbb{H}=0\}$, i.e., all the vectors that pass point $\boldsymbol{\mu}$ and are orthogonal to vector $\boldsymbol{\mu}$ based on the Lorentzian inner product. A point $(z_0,z_1,\ldots,z_M)^T$ in the Lorentz model can be conveniently mapped to the Poincaré ball[21] for visualization:

$$\left(0,\frac{(z_1,\ldots,z_M)^T}{z_0+1}\right)$$

We discard the first element as it is a constant of zero.

We used wrapped normal priors and wrapped normal posteriors defined in the Lorentz model to embed cells to a hyperbolic space[25,34,79]. A wrapped normal distribution in $\mathbb{H}^M$ is constructed by first defining a normal distribution on the

tangent space $\mathscr{T}_{\mu_0}\mathbb{H}^M$ (a Euclidean subspace in $\mathbb{R}^{M+1}$) at the origin $\mu_0 = (1, 0, \ldots, 0)^T$ of the hyperbolic space. Samples from a normal distribution on the tangent space are parallel-transported to desired locations and further projected onto the final hyperbolic space[25].

We used a set of invertible functions to transform samples from a normal distribution $\mathcal{N}(\mathbf{z}|\mathbf{0}, \mathbf{I}_M\boldsymbol{\sigma})$ in $\mathbb{R}^M$ to samples from a wrapped normal distribution in $\mathbb{H}^M$ with mean of $\boldsymbol{\mu}$, where $\boldsymbol{\sigma} \in \mathbb{R}^M$ is the standard deviation of components $z_1$ to $z_M$, respectively, and $\mathbf{I}_M$ is the identity matrix in $\mathbb{R}^{M}$[25,55]. First, let $\mathbf{z}_0 = (0, \mathbf{z}_0')^T$, which can be considered as a sample vector from $\mathscr{T}_{\mu_0}\mathbb{H}^M$, where $\mathbf{z}_0'$ is sampled from $\mathcal{N}(\mathbf{z}|\mathbf{0}, \mathbf{I}_M\boldsymbol{\sigma})$. Next, $\mathbf{z}_0$ is parallel-transported to vector $\mathbf{z}_1$ in the tangent space $\mathscr{T}_{\mu}\mathbb{H}^M$ at $\boldsymbol{\mu}$, in a parallel manner (i.e., $\mathbf{z}_1$ and $\mathbf{z}_0$ pointing in the same direction relative to the geodesic between $\mu_0$ and $\boldsymbol{\mu}$) and vector norm preserving (i.e., $\langle \mathbf{z}_0, \mathbf{z}_0 \rangle_{\mathbb{H}} = \langle \mathbf{z}_1, \mathbf{z}_1 \rangle_{\mathbb{H}}$)[25,80]:

$$\mathbf{z}_1 = \mathbf{z}_0 + \frac{\langle \boldsymbol{\mu}, \mathbf{z}_0 \rangle_{\mathbb{H}}}{\alpha + 1}(\mu_0 + \boldsymbol{\mu})$$

with $\alpha = -\langle \mu_0, \boldsymbol{\mu} \rangle_{\mathbb{H}}$.

Finally, the exponential map[24,25,79] projects vector $\mathbf{z}_1$ in the tangent space $\mathscr{T}_{\mu}\mathbb{H}^M$ back to the hyperbolic space by:

$$\mathbf{z} = \cosh(\|\mathbf{z}_1\|_{\mathbb{H}})\boldsymbol{\mu} + \sinh(\|\mathbf{z}_1\|_{\mathbb{H}})\frac{\mathbf{z}_1}{\|\mathbf{z}_1\|_{\mathbb{H}}}$$

such that the vector norm is preserved: $\|\mathbf{z}_1\|_{\mathbb{H}} = \sqrt{\langle \mathbf{z}_1, \mathbf{z}_1 \rangle_{\mathbb{H}}} = d_{\mathbb{H}}(\boldsymbol{\mu}, \mathbf{z})$.

The likelihood after the invertible transformations can be calculated by

$$\log p(\mathbf{z}) = \log p(\mathbf{z}_0) - \log\left(\det\left(\frac{\partial \mathbf{z}}{\partial \mathbf{z}_1}\right)\right) - \log\left(\det\left(\frac{\partial \mathbf{z}_1}{\partial \mathbf{z}_0}\right)\right)$$
$$= \log p(\mathbf{z}_0) - (d-1)\log\left(\frac{\sinh(\|\mathbf{z}_1\|_{\mathbb{H}})}{\|\mathbf{z}_1\|_{\mathbb{H}}}\right)$$

The encoder outputs a vector $\mathbf{h}$ in the tangent space at the origin ($\mathscr{T}_{\mu_0}\mathbb{H}^M$, so $\|\mathbf{h}\|_{\mathbb{H}} = \|\mathbf{h}\|_2$) and can be mapped to $\mathbb{H}^M$ using the exponential map (the first zero element of $\mathbf{h}$ is omitted) to get $\boldsymbol{\mu}$:

$$\boldsymbol{\mu} = \left(\cosh(\|\mathbf{h}\|_2), \sinh(\|\mathbf{h}\|_2)\frac{\mathbf{h}}{\|\mathbf{h}\|_2}\right)$$

Given a sample $\mathbf{z}$ from the wrapped normal distribution, we need to evaluate its density $\log p(\mathbf{z})$ for calculating the $\mathbb{KL}$-divergence term of the ELBO. We can use the inverse exponential map and the inverse parallel transport to compute the corresponding $\mathbf{z}_1$ and $\mathbf{z}_0$, respectively, for evaluating the density:

$$\mathbf{z}_1 = \frac{\cosh^{-1}(\beta)}{\sqrt{\beta^2 - 1}}(\mathbf{z} - \beta\boldsymbol{\mu})$$
$$\mathbf{z}_0 = \mathbf{z}_1 + \frac{\langle \mu_0, \mathbf{z}_1 \rangle_{\mathbb{H}}}{\alpha + 1}(\boldsymbol{\mu} + \mu_0)$$

where $\beta = -\langle \boldsymbol{\mu}, \mathbf{z} \rangle_{\mathbb{H}}$ and $\alpha = -\langle \mu_0, \boldsymbol{\mu} \rangle_{\mathbb{H}}$. We now have all the ingredients to compute Eq. (1) for each training point.

**Model structure**. As single-cell data are sparse, with typically >90% genes with zero counts in each cell, we used softmax as the activation function to estimate the means of the negative-binomial distributions and help generate sparse outputs from the decoders. The softmax function outputs a vector of positive numbers with a sum of one, and this vector is multiplied by the size vector of a cell (the sum of UMI counts for that cell) to get the means of the negative-binomial distribution of each gene in that cell. We used the exponential linear unit (ELU)[81] activation functions for hidden layers, as it has been shown to improve convergence of stochastic gradient optimizations.

For all experiments, we used a three-layered encoder network (128–64–32) and a two-layered decoder network (32–128). The dimensionality $\mathbf{z}$ of the stochastic layer was typically two for visualization purposes. When comparing scPhere with different latent spaces, we kept all other factors the same. We used the Adam stochastic optimization[82] algorithm with a learning rate of 0.001. For datasets with <10,000 cells, we trained models for 2000 epochs. For datasets with >10,000 cells but <100,000 cells, we trained models for 500 epochs, and for the large number of immune cells with more than 2,000,000 cells, we trained models for 250 epochs. Using the UC epithelial cells as an example, we provided the average ELBO changes with training mini-batches for different latent dimensionalities. For scPhere models with different latent spaces, training was quite stable and converged rapidly, at least for the configurations we used (Supplementary Fig. 20b). For scPhere with hyperbolic latent spaces, training converged a bit more slowly when we increased the dimensionality of the latent spaces, compared to training using other latent spaces.

For our current implementation, we did not introduce an early stop but trained scPhere for a given number of epochs. Larger datasets may require a smaller number of training epochs compared with smaller datasets (e.g., 1000 cells). Training time also depended on the number of genes used. For example, when using the IBD immune cells, time grew linearly with the number of mini-batches in training, taking only 2.45 min to train scPhere for 16,450 mini-batches

(Supplementary Fig. 4a). We can estimate the number of cells equivalent with the mini-batch size (128) and number of mini-batches in training (Supplementary Fig. 4b). Importantly, we obtained good embedding for the 210,614 immune cells, even when we only train the model for ten epochs (16,450 mini-batches, 2.45 min, Supplementary Fig. 4c–i). All the experiments were run using a Mac desktop computer with 32 GB of RAM, 4.2 GHz four-core Intel i7 processor with 8 MB cache and no GPUs were used.

**Parameter setting for other methods used in comparisons**. For t-SNE, we followed the previous approach optimized for visualizing scRNA-seq data[18], i.e., using PCA initiation, a high learning rate, and multi-scale similarity kernels. We also used the Fit-SNE package[83], as previously described[18].

For UMAP, we used the Seurat UMAP wrapper[5] and adapted its parameter setting to run UMAP, with the "min.dist" parameter set to 0.3 and the "spread" parameter set to 1.

For PHATE[37], we followed its tutorial and used the default parameter settings.

For these three methods, we used 50 principal components as inputs by default. Because PHATE had very long run times for large datasets, for the IBD immune cells we only ran PHATE with 2D latent spaces.

For the three batch-correction methods, Harmony[30], Seurat3 CCA[5], and LIGER[29], we followed their tutorials and used the default parameter settings. Because both Seurat3 CCA and LIGER only handle one batch vector, we used patient, which is the major batch factor for the UC data. Seurat3 CCA encountered scalability/stability issue for the immune cells with >200,000 cells and 30 batches (patients) and failed after running >90 h. We thus removed Seurat3 CCA from the immune cell comparison (Supplementary Fig. 8).

**Quantifying global, hierarchical structure preservation**. The preservation of global hierarchical structures for each embedding method was quantified by using a "global k-NN accuracy" metric, where k-NN classifiers (leave-one-out cross-validation, $k = 3$ and 5 for the RGC and HCL datasets, respectively) were trained on condensed datasets, where each point was the center of a cluster, and the classifiers were used to classify each cluster, which was represented by its cluster center, to the major cell types (groups).

To quantify cell crowding, we used silhouette values. When cells are more crowded, within-cluster and between-cluster distances between cells are more similar to each other, leading to smaller silhouette values. For the HCL dataset with 599,926 cells, silhouette values were calculated from 50 repeated runs, each run with 20,000 randomly sampled cells as inputs.

The embeddings were visualized on a 3D sphere using the rgl package[84] from R, with the interactive 3D scatter plots saved as web graphics library files that can be opened in a browser. The rgl package uses OpenGL as the rendering backend and can be used to rapidly and interactively visualize 3D scatter plots with millions of cells in a browser.

To learn scPhere models that are invariant to the batch vectors and can be used to map cells from completely new batches, when training scPhere, we use a scPhere encoder (the encoder part of the scPhere model is used to map new data after a scPhere model is trained) to map a gene expression vector to the low-dimensional representation directly without using the batch vector as an input to the encoder. The batch vector is only used in the decoder that took both the latent representation of a cell and its cell batch vector as input to output the recovered gene expression vector during training scPhere. We call this modality of scPhere with no batch vectors for the encoder "batch-invariant" scPhere, as it learns latent representations that are invariant to the batch vectors.

**The component-collapse problems in VAE**. We examined if scPhere with a high-dimensional latent variable ($\mathbf{z}$) has the "component-collapse" problem[33], where the generative model (decoder) simply ignores some components of the latent variables, such that the posteriors of these components match the prior.

For Euclidean latent spaces, we observed the component-collapse problem when we used either 10D or 20D, where the means of the absolute values of some components were close to zero—the mean of the standard normal distribution (Supplementary Fig. 10a). Therefore, the effective numbers of components by using 10D and 20D dimensional latent spaces were only six and seven, respectively.

For hyperspherical latent spaces, because the prior has no centers and all components shared the same concentration parameter, we did not observe component-collapse, so potentially we can get a larger number of effective latent components compared with using Euclidean latent spaces. However, when using 20D hyperspherical latent spaces, the estimated vMF concentration parameters were lower compared to the case with latent variables on 5-spheres (Supplementary Fig. 10b), suggesting higher uncertainties of the embedding when using latent variables on a 20-sphere. Moreover, some components of the latent representations became highly correlated when we embedded cells on a 20-sphere (Supplementary Fig. 10c). By using the hyperbolic latent spaces, we also observed co-linear components with 5D-latent spaces.

**Datasets**. The cord blood mononuclear cell dataset[27] consists of 8617 cells, including 8009 cord blood mononuclear cells and 608 mouse 3T3 fibroblasts, produced by the CITE-seq protocol[38] on the 10× Chromium (v2) platform[85]. We only used the 2293

$CD14^+$ erythroid cells and the first 10 erythrocytes in the dataset. Based on the Seurat[5] tutorial (https://satijalab.org/seurat/v3.0/multimodal_vignette.html), we used the 2000 highly variable genes in this study.

Human splenic NK cells were from a study profiling human and mouse splenic and blood NK cells[41], and profiled by 10× Chromium (v2). We used the 1755 human splenic NK cells from donor one in this study. We selected 2724 highly variable genes and partitioned the 1755 cells into four groups, labeled them as hNK_Sp1, hNK_Sp2, hNK_Sp3, hNK_Sp4, as in the original study[41].

Human lung cells were from asthma patients and healthy controls[39], and profiled by either 10× Chromium or Drop-seq[86]. We used the 3314 cells from a donor prepared by the Drop-seq protocol that can be accessed from GEO: GSE130148.

The mouse white adipose tissue stromal cell dataset contains 1378 cells from mouse white adipose tissue[40] profiled by 10× Chromium (v2). In the original study, the authors only analyzed 1045 tdTomato- mGFP+ cells and identified adipocyte precursor cells (APC), fibro-inflammatory progenitors (FIP), committed preadipocytes, and mesothelial cells. We analyzed all the cells and further identified pericytes, macrophages, and two groups of doublets.

The reginal ganglion cell atlas dataset consists of 35,699 mouse retinal ganglion cells profiled by 10× Chromium (v2)[42]. The original analysis identified 45 clusters, and one cluster consisted of two cell types.

We used 599,926 human cell landscape cells[43] from human fetal or adult tissues profiled by the Microwell-Seq platform[87]. These cells were portioned into 102 clusters, and 77 of 102-cell clusters can be grouped into six major cell groups: fetal stromal cells, fetal epithelial cells, adult endothelial cells, endothelial cells, adult stromal cells, and immune cells.

The cells of the colon mucosa were from 68 biopsies, collected from 18 ulcerative colitis patients and 12 healthy individuals[44], and profiled by 10× Chromium (either v1 or v2). After filtering likely low-quality cells (clusters), we obtained a total of 301,749 cells (26,678 stromal cells and glia, 64,457 epithelial cells, and 210,614 immune cells as annotated in the original study[44]). The cells span 12 stromal cell types/states, 12 epithelial cell types/states, and 23 immune cell types/states, identified by unsupervised clustering and manual annotations[44]. We used Seurat to select 1307, 1361, and 1068 highly variable genes for the three major cell types, respectively, for scPhere analyses.

The C. elegans embryonic cell dataset consists of 86,024 cells[51] profiled using 10× Chromium (v2). The embryo times were partitioned into 12-time bins, and 63.5% of the cells were assigned to 36 major cell types based on annotation from GEO: GSE126954. We treated the cells with embryo time in the range 100–130 as the root cells because embryo time <100 consisted mostly of germline cells that were also observed in embryo time >650. As the root cells accounted for only ~0.5% of the total cells, it was hard for them to be mapped to the center of the Lorentz model. We thus changed the scPhere objective function by adding a term consisting of the distance between the mean position of embeddings of the root cells to the origin of the Lorentz model.

For the zebrafish embryonic cells, we used the 5226 cells at 50% epiboly[48], profiled using Drop-seq[86]. These cells were from four batches, and we used the 1406 cells from one batch to train scPhere, and the trained model was used to map the remaining 3820 cells from another three batches. We used Seurat to select 2000 variable genes, added three genes with annotated spatial locations that were not in the 2000 variable gene list, and used the 2003 genes for analysis. To encourage cells to map to their spatial locations, we used 11 landmark genes expressed in the ventral axis: cdx4, eve1; animal ventral: bambia; dorsal: gsc, chd; animal dorsal: foxd3; marginal: osr1, lft2, lhx1a, wnt8a; and the gene ta which is expressed in ventral, dorsal, and marginal. These landmark genes were used to calculate a penalty term added to the scPhere objective function during training.

**Reporting summary**. Further information on research design is available in the Nature Research Reporting Summary linked to this article.

## Data availability

We used publicly available datasets in this study (GEO: GSE126954[38], GSE119562[41], GSE130148[39], GSE111588[40], GSE137400[42], GSE134355[43], GSE126954[51], GSE106587[48]; Single Cell Portal: SCP259. To make the results presented in this study reproducible, all processed data are available in the Single Cell Portal (SCP551).

## Code availability

The scPhere software package, implemented in TensorFlow, is available free from https://github.com/klarman-cell-observatory/scPhere, and as a Supplementary Software 1 accompanying this manuscript.

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

## Acknowledgements

We thank Jennifer Rood for helpful comments and Leslie Gaffney for help with figure preparation, Inbal Benhar and Karthik Shekhar for the RGC data analysis, Jeffrey A Farrell for mapping zebrafish embryonic cells. This work was supported by the Klarman Cell Observatory, HHMI, the Food Allergy Science Initiative, the Manton Foundation, the NIH BRAIN Initiative (1U19 MH114821), and NIH/National Institute of Diabetes and Digestive and Kidney Diseases grant (1RC2DK114784).

## Author contributions

J.D. and A.R. developed the model. J.D. conducted experimental analyses with guidance from A.R. J.D., and A.R. interpreted the results and wrote the manuscript.

## Competing interests

A.R. is a founder and equity holder of Celsius Therapeutics, an equity holder in Immunitas Therapeutics and until August 31, 2020 was a SAB member of Syros Pharmaceuticals, Neogene Therapeutics, Asimov, and ThermoFisher Scientific. From August 1, 2020, A.R. is an employee of Genentech, a member of the Roche Group. J.D. declares no competing interests.
