## [Peer Review File · Nature Communications]

Reviewers' Comments:

Reviewer #1:

None

Reviewer #2:

Remarks to the Author:

Ding and Regev have developed scSphere, a variational autoencoder for embedding scRNA data into non-Euclidean latent spaces, including hyperspheres and hyperbolic spaces. The authors have done an admirable job of responding to the previous reviewer comments. The quantitative comparisons with previous methods are thorough. It appears that the method provides advantages over VAEs, t-SNE, and UMAP in visualization and over existing methods for batch correction. Furthermore, the method itself is interesting and seems technically sound. I have no concerns that would require major revisions or prohibit publication.

Reviewer #3:

Remarks to the Author:

The authors have satisfactorily addressed my questions. Thank you.

REVIEWERS' COMMENTS

Reviewer #2 (Remarks to the Author):

Ding and Regev have developed scPhere, a variational autoencoder for embedding scRNA data into non-Euclidean latent spaces, including hyperspheres and hyperbolic spaces. The authors have done an admirable job of responding to the previous reviewer comments. The quantitative comparisons with previous methods are thorough. It appears that the method provides advantages over VAEs, t-SNE, and UMAP in visualization and over existing methods for batch correction. Furthermore, the method itself is interesting and seems technically sound. I have no concerns that would require major revisions or prohibit publication.

We thank the reviewer for the thoughtful points that help improve the quality of our work.

Reviewer #3 (Remarks to the Author):

The authors have satisfactorily addressed my questions. Thank you.

We thank the reviewer for the concerns for us to think more deeply about our work.